# Cross-species functional diversity within the PIN auxin efflux protein family

**Devin Lee O'Connor[1]\*, Samuel Elton[1†], Fabrizio Ticchiarelli[1†], Mon Mandy Hsia[2], John P Vogel[3,4], Ottoline Leyser[1]**

[1]The Sainsbury Laboratory, University of Cambridge, Cambridge, United Kingdom; [2]Western Regional Research Center, USDA-ARS , Albany, United States; [3]United States Department of Energy Joint Genome Institute, Walnut Creek, United States; [4]Department of Plant and Microbial Biology, University of California, Berkeley, Berkeley, United States

**Abstract** In Arabidopsis, development during flowering is coordinated by transport of the hormone auxin mediated by polar-localized PIN-FORMED1 (AtPIN1). However Arabidopsis has lost a PIN clade sister to AtPIN1, Sister-of-PIN1 (SoPIN1), which is conserved in flowering plants. We previously proposed that the AtPIN1 organ initiation and vein patterning functions are split between the SoPIN1 and PIN1 clades in grasses. Here we show that in the grass Brachypodium *sopin1* mutants have organ initiation defects similar to Arabidopsis *atpin1*, while loss of *PIN1* function in Brachypodium has little effect on organ initiation but alters stem growth. Heterologous expression of Brachypodium SoPIN1 and PIN1b in Arabidopsis provides further evidence of functional specificity. SoPIN1 but not PIN1b can mediate flower formation in null *atpin1* mutants, although both can complement a missense allele. The behavior of SoPIN1 and PIN1b in Arabidopsis illustrates how membrane and tissue-level accumulation, transport activity, and interaction contribute to PIN functional specificity.
DOI: https://doi.org/10.7554/eLife.31804.001

**\*For correspondence:**
devin.oconnor@slcu.cam.ac.uk

[†]These authors contributed equally to this work

**Competing interests:** The authors declare that no competing interests exist.

## Introduction

The plant hormone auxin is an essential mobile signal controlling growth and patterning throughout plant development (*Leyser, 2010*). Auxin can passively enter cells triggering a vast array of downstream signaling events (*Wang and Estelle, 2014*), but it cannot easily exit the cell without active transport (*Raven, 1975*; *Rubery and Sheldrake, 1974*). As a result, directional efflux mediated by the polar-localized PIN-FORMED (PIN) efflux carriers can organize auxin flows and accumulation patterns, creating concentration maxima and paths of transport that regulate growth, position organs and pattern tissues (*Adamowski and Friml, 2015*). Because auxin itself feeds back to regulate PIN-mediated transport both transcriptionally and post-transcriptionally (*Leyser, 2006*), the transport system shows remarkable robustness and plasticity. For example compensatory changes in PIN abundance between PIN family members can mitigate PIN loss-of-function mutant phenotypes (*Blilou et al., 2005*; *Paponov et al., 2005*; *Vieten et al., 2005*), environmental inputs can trigger tissue-level changes in PIN abundance and polarity leading to altered plant growth (*Habets and Offringa, 2014*), and auxin transport paths can be reorganized in response to injury (*Sauer et al., 2006*; *Xu et al., 2006*) or spontaneously in tissue culture (*Gordon et al., 2007*). The self-organizing properties of the auxin transport system gives this patterning mechanism extraordinary versatility and allows it to coordinate both local and long-range communication in plants.

The correct initiation and positioning of organs (leaves, flowers, stems) in the growing tip, or shoot apical meristem, of *Arabidopsis thaliana* (Arabidopsis) plants requires the action of the PIN-FORMED1 (AtPIN1) auxin efflux carrier (*Okada et al., 1991*). AtPIN1 is targeted to the plasma

membrane and polarized in cells (*Gälweiler et al., 1998*). In the meristem epidermis polarization of AtPIN1 in neighboring cells converges around the initiation sites of new organs, suggesting that polarized AtPIN1 concentrates auxin into local maxima causing organ initiation (*Benková et al., 2003*; *Heisler et al., 2005*; *Reinhardt et al., 2003*). Accordingly in *atpin1* loss-of-function mutants or if auxin transport is pharmacologically inhibited, organ initiation is aborted but it can be rescued with local auxin application to the meristem flank (*Reinhardt et al., 2003*; *Reinhardt et al., 2000*). Organ initiation in *atpin1* mutants can also be rescued with epidermal-specific AtPIN1 expression (*Bilsborough et al., 2011*) and reducing AtPIN1 function specifically in the epidermis compromises organ positioning and initiation (*Kierzkowski et al., 2013*), demonstrating the importance of convergent AtPIN1 polarization in the epidermis during organ formation.

The recurrent formation of AtPIN1 convergence points surrounding auxin maxima in the meristem epidermis has been the focus of several computational models that attempt to explain how auxin feeds back on its own transport via AtPIN1 to concentrate auxin and control organ spacing (*Abley et al., 2016*; *Bayer et al., 2009*; *Bhatia et al., 2016*; *Heisler et al., 2010*; *Jönsson et al., 2006*; *Smith et al., 2006*; *Stoma et al., 2008*). However AtPIN1 is also expressed during the patterning of the vascular strands formed coincident with organ positioning, and in these sub-epidermal cells AtPIN1 is polarized rootward away from the presumed auxin maxima, suggesting that AtPIN1 polarity with respect to auxin concentration may vary across tissues or over developmental time (*Bayer et al., 2009*).

Indeed AtPIN1 has several functions post organ initiation that are not necessarily associated with convergent polarization patterns (*Gälweiler et al., 1998*; *Scarpella et al., 2006*). AtPIN1 is not required for organ formation during the vegetative phase. Mutants lacking AtPIN1 form leaves but they are misplaced and have severe morphological and vascular defects similar to those observed upon pharmacological inhibition of auxin transport, suggesting an important role for AtPIN1 in post-initiation morphogenesis and vein patterning in leaves (*Guenot et al., 2012*; *Sawchuk et al., 2013*; *Verna et al., 2015*). Furthermore in mature tissues AtPIN1 is polarized rootward in vascular-associated cells and is required for efficient long distance transport of auxin down the shoot in the polar auxin transport stream and this has been proposed to play an important role in the regulation of shoot branching (*Bennett et al., 2016*; *2006*; *Gälweiler et al., 1998*; *Shinohara et al., 2013*). Mutations in other PIN family members in combination with *atpin1* mutants suggest further functions in embryo development, root development and during plant growth responses to light and gravity (*Leyser, 2005*). Unfortunately the myriad roles for AtPIN1 during inflorescence development are genetically obscured by the severity of *atpin1* organ initiation defects.

We previously showed that all sampled flowering plants outside of the Brassicacea family have a clade of PIN proteins phylogenetically sister to the PIN1 clade (The Sister-of-PIN1 or SoPIN1 clade), while Arabidopsis and other Brassicacea species have lost this clade (*O'Connor et al., 2014*). During initiation of the lemma, a leaf-like floral organ in the grass *Brachypodium distachyon* (Brachypodium), which has both PIN1 and SoPIN1 clades, SoPIN1 is highly expressed in the epidermis, polarizes towards presumed auxin maxima, and forms convergent polarization patterns, suggesting a role in creating the auxin maxima required for organ initiation in the shoot. In contrast, the duplicate Brachypodium PIN1 clade members, PIN1a and PIN1b, are not highly expressed in the epidermis, orient away from presumed auxin maxima and are primarily expressed during patterning in the sub-epidermal tissues. Thus the combined expression domains and polarization behaviors of SoPIN1, PIN1a, and PIN1b in Brachypodium largely recapitulate those observed for AtPIN1 alone in Arabidopsis.

The dynamic localization and polarization patterns of the Brachypodium SoPIN1 and PIN1 clades can be modeled with two different polarization modes with respect to auxin. PIN behaviors can be captured by a model in which SoPIN1 polarizes 'up-the-gradient' towards the neighboring cell with the highest auxin concentration, while PIN1a and PIN1b polarize 'with-the-flux' accumulating in the membrane with the highest net auxin efflux (*O'Connor et al., 2014*). Both polarization modes were previously applied to AtPIN1 in order to capture the switch in polarity observed during organ initiation and vein patterning, first orienting toward the auxin maximum during convergence point formation, then orienting away from the maximum below the epidermis during vein patterning (*Bayer et al., 2009*). These localization and modeling results suggest that in most angiosperm species the organ placement and vascular patterning functions attributed to AtPIN1 in Arabidopsis are split between the PIN1 and SoPIN1 clades and that these two clades have different polarization properties with respect to auxin.

Exploring this hypothesis, here we present the functional analysis of both the SoPIN1 and PIN1 protein clades in Brachypodium, a species with the canonical two-clade family structure. We show that SoPIN1 and the PIN1 clade members PIN1a and PIN1b have different functions during Brachypodium development, with SoPIN1 being required for organ initiation during the flowering phase, and PIN1a and PIN1b regulating stem growth. Using heterologous expression in Arabidopsis we show that the SoPIN1 and PIN1b proteins have different accumulation, polarization, and transport behaviors that result in different functional properties independent of transcriptional context. In addition to elucidating several ways in which PIN family members can be functionally distinct, these results suggest that the Arabidopsis AtPIN1 protein represents an example of an evolutionary phenomenon the opposite of subfunctionalisation in which protein functions are amalgamated into a single protein rather than diversified amongst paralogs. AtPIN1 has a repertoire of roles and associated polarization behaviors that are distributed among several clades of PIN proteins in most flowering plants.

## Results

### The SoPIN1 and PIN1 clades have different functions in Brachypodium

We targeted Brachypodium *SoPIN1*, *PIN1a* and *PIN1b* with gene-specific Clustered Regularly Interspaced Short Palindromic Repeats (CRISPR) and for all three genes recovered independent single base-pair lesions causing frame shifts and premature stop codons (*Figure 1A*). The wild-type Brachypodium inflorescence meristem normally makes several lateral spikelet meristems (lsm) before producing a terminal spikelet meristem (tsm) (*Figure 1B*)(*Derbyshire and Byrne, 2013*). Both lateral and terminal spikelet meristems are consumed in the production of florets (*Figure 1D and F*). The *sopin1-1* inflorescence meristems had severe organ and spikelet branch initiation defects (*Figure 1C*), which resulted in reduced total whole-plant spikelet number (*Figure 1H*). When spikelets did form, *sopin1-1* spikelet meristems were often devoid of new organs (*Figure 1E*) and very few recognizable florets were produced (*Figure 1I*). In support of the *sopin1-1* lesion being responsible for these varied inflorescence phenotypes, we complemented inflorescence development and seed set by crossing *sopin1-1* to the previously published SoPIN1-Citrine fusion line (*Figure 1—figure supplement 1*)(*O'Connor et al., 2014*). The pleotropic defects displayed by *sopin1-1* in the inflorescence are remarkably similar to loss-of-function *pin1* mutants in *Arabidopsis* (*Okada et al., 1991*) and *Cardamine hirsuta* (*Barkoulas et al., 2008*).

In wild-type spikelet meristems SoPIN1 convergence point formation is coincident with an increase in the auxin signaling reporter DR5 (*O'Connor et al., 2014*), as well as a decrease in the nuclear auxin response reporter protein DII-Venus (*Brunoud et al., 2012*) (DII) (*Figure 1J*), which functions in Brachypodium and is degraded in the presence of auxin in spikelet meristems (*Figure 1—figure supplement 2*). In *sopin1-1* meristems DII accumulation was uniformly high for long stretches of the epidermis and the patterned reduction of DII both in the meristem epidermis and internally failed to occur, suggesting a failure to organize auxin maxima (*Figure 1K* arrow).

In contrast to the severe defects of *sopin1-1*, organ initiation in *pin1a-1* and *pin1b-1* single mutants was largely unaffected. The mature inflorescences of both *pin1a-1* and *pin1b-1* had normal spikelets (*Figure 1F*) and spikelet meristem morphology was indistinguishable from wild-type (*Figure 1G*). Mutant *pin1a-1* plants appeared visually wild-type but we measured a slight increase in total spikelet number (*Figure 1H*). Mutant *pin1b-1* plants were similar to wild-type with respect to both spikelet and floret numbers (*Figure 1H and I*) but often had bent apical internodes (*Figure 1F* arrowhead). While *pin1a-1* and *pin1b-1* single mutants had no clear organ initiation defects they showed changes in internode length (*Figure 2*). Plant stature in *pin1a-1* mutants was largely indistinguishable from wild-type (*Figure 2B*) but we measured a small reduction in the length of the I4 internode (*Figure 2E*). In contrast *pin1b-1* plants were easily distinguished from wild-type because of a significant increase in internode length at the base of the plant, resulting in greater overall plant height (*Figure 2E*). The elongated basal internodes and bent stems of *pin1b-1* resulted in a less compact plant architecture compared to the other genotypes (*Figure 2C*). The increase in basal internode length in *pin1b-1* single mutants was rescued by the previously published PIN1b-Citrine fluorescent reporter (*O'Connor et al., 2014*) (*Figure 2—figure supplement 1*).

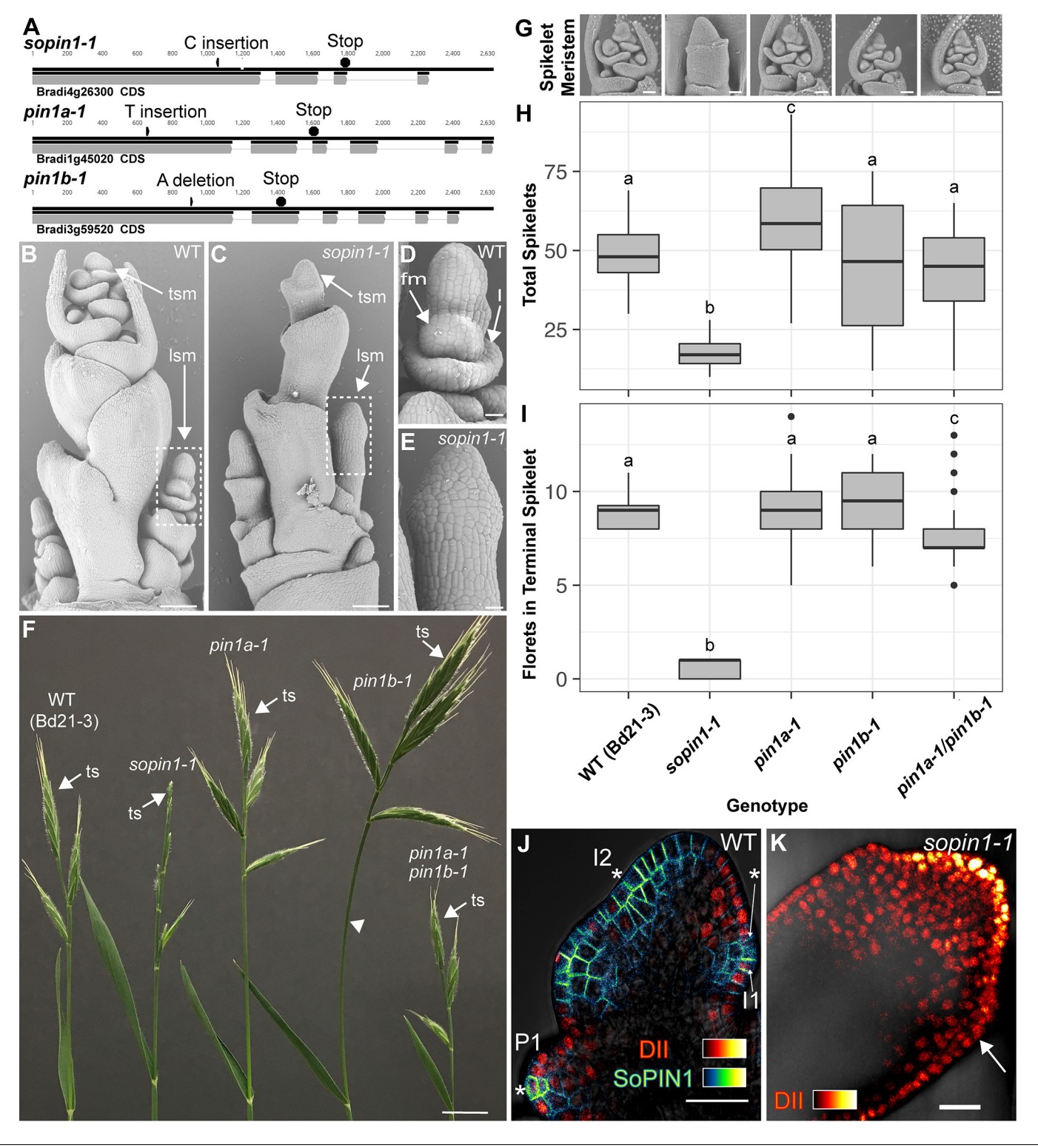

**Figure 1.** Mutation of SoPIN1 but not PIN1a and PIN1b severely effects organ initiation in Brachypodium. (**A**) *SoPIN1*, *PIN1a*, and *PIN1b* CRISPR-derived mutant alleles (see Materials and methods). Coding sequences are indicated by grey boxes. Arrowheads indicate CRISPR target sites and are labeled with the type of DNA lesion (C insertion, T insertion, or A deletion). All mutant alleles have frame shifts that result in premature stop codons at the positions indicated. (**B–G**) Inflorescence phenotypes of CRISPR-derived *sopin1-1, pin1a-1,* and *pin1b-1* mutants. See *Figure 2* for whole-plant phenotypes. (**B–E**) and (**G**) are scanning electron microscopy (SEM). (**B**) Immature wild-type (WT) (inbred line Bd21-3) Brachypodium inflorescence with several lateral spikelet meristems (lsm), and a terminal spikelet meristem (tsm). (**C**) *sopin1-1* plants have severe organ initiation defects in the

*Figure 1 continued on next page*

*Figure 1 continued*

inflorescence. (D) Detail of a wild-type lateral spikelet meristem outlined by a box in (B) showing an immature lemma (I), which is the leaf-like organ that subtends the floral meristem (fm). (E) Detail of barren lateral spikelet meristem outlined by box in (C). (F) Mature inflorescence phenotypes of WT (Inbred Bd21-3), *sopin1-1*, *pin1a-1*, *pin1b-1*, and double *pin1a-1/pin1b-1* mutants. The terminal spikelet (ts) of each inflorescence is indicated for comparison. Arrowhead indicates bent internode tissue in *pin1b-1*. Genotypes for (G–I) are indicated at the bottom of (I). (G) SEM details of representative spikelet meristems. (H) Box-plot of total whole-plant spikelet number at seed-set. (n = 22–53 plants each genotype). Samples with different letters are significantly different from each other (ANOVA, Tukey HSD, p<0.05). See '*Figure 1H–I* Source Data 1' for source data. (I) Box-plot of the number of florets in each terminal spikelet of the central branch at seed set. (n = 22–53 plants each genotype). Samples with different letters are significantly different from each other (ANOVA, Tukey HSD, p<0.05). See '*Figure 1H–I* Source Data 1' for source data. (J) Medial confocal Z-section of pZmUbi::DII-Venus (DII) expression in a WT spikelet co-expressing SoPIN1 tagged with Cerulean (a CFP variant) under the native *SoPIN1* promoter. Organ primordia are numbered I2, I1, P1 from youngest to oldest. DII is normally degraded at SoPIN1 convergence points in I2 and I1 primordia (asterisks), and in response to auxin treatment (See *Figure 1—figure supplement 2*). Inset shows color look-up-table for all subsequent PIN images and color look-up-table for DII. (K) Medial confocal Z-section of pZmUbi::DII-Venus expression in a *sopin1-1* spikelet meristem. DII degradation does not occur in the periphery of *sopin1-1* meristems, and organs fail to initiate (arrow). Scale bars: 100 µm in (B) and (C), 20 µm in (D) and (E), 1 cm in (F), 50 µm each in (G), and 25 µm in (J) and (K).

DOI: https://doi.org/10.7554/eLife.31804.002

The following source data and figure supplements are available for figure 1:

**Source data 1.** Source data for spikelet and floret counts in *Figure 1H–I*.
DOI: https://doi.org/10.7554/eLife.31804.005
**Source data 2.** Source data for DII quantification in *Figure 1—figure supplement 2* panel C.
DOI: https://doi.org/10.7554/eLife.31804.006
**Figure supplement 1.** *sopin1-1* is complemented by the SoPIN1-CIT reporter.
DOI: https://doi.org/10.7554/eLife.31804.003
**Figure supplement 2.** DII-Venus is degraded in the presence of auxin in Brachypodium spikelet meristems.
DOI: https://doi.org/10.7554/eLife.31804.004

The PIN1a and PIN1b duplication is specific to, but conserved within the grasses (*O'Connor et al., 2014*). Thus we suspected these two genes would show a degree of genetic redundancy. Indeed *pin1a-1/pin1b-1* (*pin1a/b*) double mutants showed a synergistic phenotype with severely reduced plant height (*Figure 2D*) resulting primarily from reduced internode growth in the upper internodes (*Figure 2E*). However, despite loss of both PIN1a and PIN1b function, *pin1a/b* double mutants made normal spikelet meristems (*Figure 1G*), had a wild-type total spikelet number (*Figure 1H*) and showed only a small reduction in floret number in the terminal spikelet (*Figure 1I*). In addition unlike *sopin1-1* plants, *pin1a/b* double mutants set ample seed.

Combined these phenotypes provide further support for functional distinction between the SoPIN1 and PIN1 clades and indicate that while the PIN1 clade is expendable for organ initiation in Brachypodium, it is involved in the regulation of internode growth.

## AtPIN1, SoPIN1 and PIN1b accumulate differently in Arabidopsis under the same transcriptional control

During organ formation in the Brachypodium shoot, expression of both SoPIN1 and PIN1b precedes PIN1a, which only accumulates significantly at the sites of vein formation after organs begin to grow (*O'Connor et al., 2014*). In the earliest stages of initiation prior to the periclinal cell divisions that are the hallmark of morphogenesis, SoPIN1 forms convergent polarization patterns around presumed auxin maxima in the meristem epidermis, while PIN1b is expressed internally and orients away from maxima (*O'Connor et al., 2014*). Because of their early expression, opposing polarization patterns and their clear single-mutant phenotypes in Brachypodium, we focused on characterizing SoPIN1 and PIN1b as representatives of the SoPIN1 and PIN1 clades.

The difference between the *sopin1-1* and *pin1b-1* phenotypes in Brachypodium could be due to their different expression patterns and not necessarily to differences in their polarization with respect to auxin concentration or flux as previously hypothesized (*O'Connor et al., 2014*). In order to assess the functional differences between the proteins independent of transcriptional context, we expressed both Brachypodium proteins tagged with Citrine (a YFP derivative) in wild-type Arabidopsis (Columbia, Col-0) under the control of a 3.5 kb Arabidopsis *PIN1* promoter fragment which includes sequences known to drive PIN1 expression sufficient to complement *pin1* mutants (*proAtPIN1*) (*Benková et al., 2003*; *Heisler et al., 2005*). In the Arabidopsis inflorescence meristem, wild-

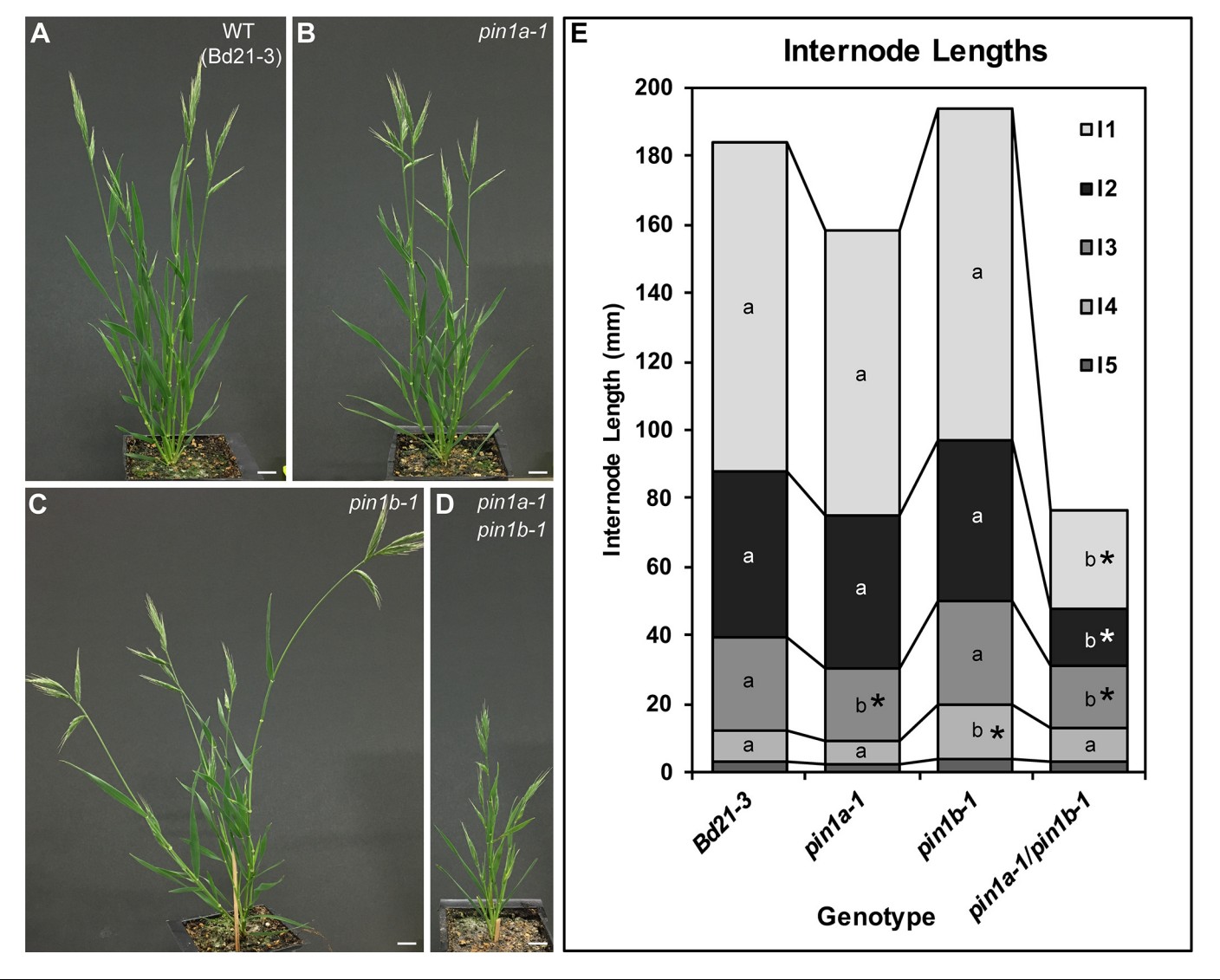

**Figure 2.** PIN1a and PIN1b redundantly control internode growth in Brachypodium. (A–D) Whole-plant phenotypes for WT (Bd21-3), *pin1a-1*, *pin1b-1*, and double *pin1a-1/pin1b-1* mutants. (E) Stacked bar graph of the length of the first 5 internodes below the inflorescence of the main branch, labeled I1-I5 from top to bottom. Lines connect analogous internodes between genotypes. Analogous internodes with different letters are significantly different from each other (ANOVA, Tukey HSD, p<0.05). I5 internodes were not significantly different between genotypes and are unlabeled. Internode lengths significantly different from WT are indicated by asterisks. (n = 18–51 individuals each genotype) See '*Figure 2—source data 1*' for source data. Scale bars: 1 cm in (A–D).

DOI: https://doi.org/10.7554/eLife.31804.007

The following source data and figure supplement are available for figure 2:

**Source data 1.** Source data for internode length measurements in *Figure 2E*.
DOI: https://doi.org/10.7554/eLife.31804.009

**Source data 2.** Source data for PIN1b-CIT-mediated complementation of *pin1b-1* internode lengths in *Figure 2—figure supplement 1*.
DOI: https://doi.org/10.7554/eLife.31804.010

**Figure supplement 1.** PIN1b-CIT-mediated complementation of *pin1b-1* internode length defects.
DOI: https://doi.org/10.7554/eLife.31804.008

type AtPIN1 forms convergent polarization patterns that mark the sites of initiating flower primordia (*Figure 3A*). Remarkably, despite the loss of the SoPIN1 clade from Arabidopsis, Brachypodium SoPIN1 also created clear convergent polarization patterns in Arabidopsis inflorescence meristems but was less abundant in the central domain of the apical dome (*Figures 3B*, 25 of 27 meristems

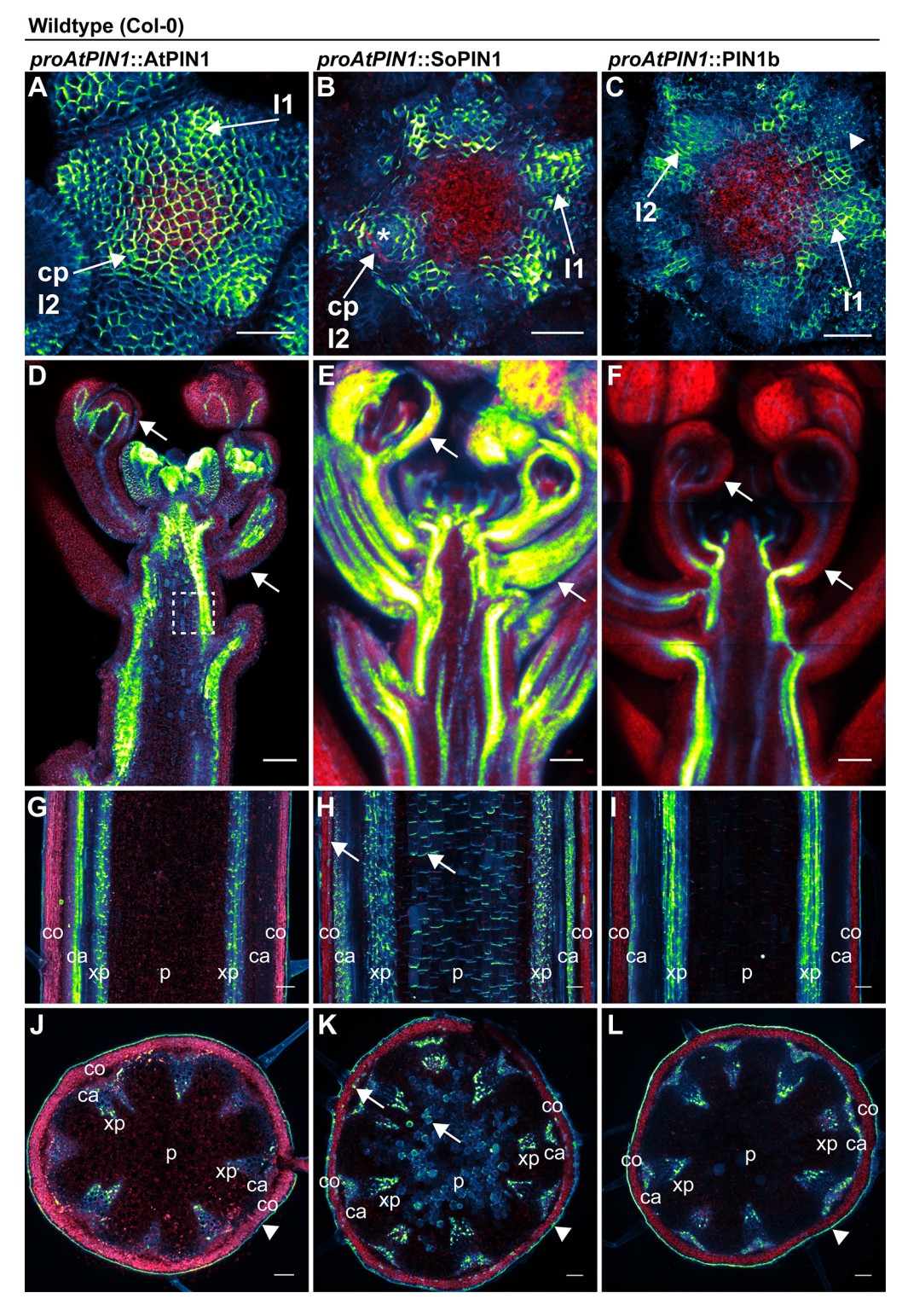

**Figure 3.** AtPIN1, SoPIN1, and PIN1b show different behaviors when expressed in wild-type Arabidopsis. Arabidopsis *AtPIN1* promoter (*proAtPIN1*) driven expression of GFP-tagged AtPIN1 and Citrine-tagged (a YFP derivative) SoPIN1 and PIN1b in wild-type Columbia (Col-0) Arabidopsis. (**A,D,G,J**) AtPIN1, (**B,E,H,K**) SoPIN1, (**C,F,I,L**) PIN1b. (**A–C**) Maximum projections of inflorescence meristem apexes. Arrows in (**A**) and (**B**) indicate convergence points (cp) in I2 primordium. Arrowhead in (**C**) indicates internalized PIN1b in punctate membrane

*Figure 3 continued on next page*

*Figure 3 continued*

bodies. The I2 and I1 primordia are labeled. (**D–F**) Tiled confocal maximum projections of longitudinal hand-sections through inflorescence apexes. Arrows indicate SoPIN1 epidermal accumulation in sepal primordia and flower pedicels in (**E**) and the lack of AtPIN1 and PIN1b epidermal accumulation in the same tissues in (**D**) and (**F**). Box in (**D**) shows detail area in *Figure 3—figure supplement 1* panel D. (**G–I**) Tiled confocal maximum projections of longitudinal hand-sections through mature basal inflorescence stem internodes 1 cm above the rosette. (**J–L**) Tiled confocal maximum projections of hand cross-sections through basal internodes 1 cm above the rosette. Signal at the edge of each section (arrowheads) is cuticle auto-fluorescence. The cortex (co), cambium (ca), xylem parenchyma (xp), and pith (p) tissues are indicated in (**G–L**). Arrows in (**H**) and (**K**) indicate cortex and pith ectopic accumulation of SoPIN1. Red signal in all panels is chlorophyll auto-florescence. Scale bars: 25 μm in (**A–C**), and 100 μm in (**D–L**).

DOI: https://doi.org/10.7554/eLife.31804.011

The following figure supplement is available for figure 3:

**Figure supplement 1.** *proAtPIN1* AtPIN1, SoPIN1, and PIN1b expression details.

DOI: https://doi.org/10.7554/eLife.31804.012

from 4 independent transgenic events). Similar to AtPIN1, SoPIN1 protein abundance was highest in the meristem epidermis and SoPIN1 convergence points were most clearly observed surrounding I2 and I1 primordia (*Figure 3B*). Below the epidermis wild-type AtPIN1 accumulates in small groups of cells that will become the vasculature (*Figure 3—figure supplement 1* panel A arrows). In contrast, sub-epidermal SoPIN1 accumulated in an ill-defined ring shape surrounding the meristem central domain without distinct foci of accumulation (*Figure 3—figure supplement 1* panel B, 15 of 23 meristems from 4 independent transgenic events).

In contrast to both AtPIN1 and SoPIN1, under the same promoter significant PIN1b accumulation was absent from the meristem epidermis in 19 of 29 meristems from 7 independent transgenic events. In the meristems where PIN1b accumulated in the epidermis it did not show clear convergent polarization patterns and its polarity was often unclear (*Figures 3C*, 10 meristems from 2 events). Within initiating organs PIN1b often localized to punctate vesicular bodies inside cells, not in the cell membrane (*Figure 3C* arrowhead). PIN1b accumulation remained low just below the meristem apex, but in contrast to SoPIN1, PIN1b formed defined domains around the presumptive developing vascular bundles similar to AtPIN1 (*Figure 3—figure supplement 1* panel C arrows). The lack of PIN1b protein in the meristem epidermis was not due to silencing of the transgene in these lines because we observed abundant PIN1b protein in the developing vasculature below the apex, even in plants where the meristem had no detectable epidermal expression (*Figure 3F*) (8 samples from 4 events). In contrast, AtPIN1 and SoPIN1 accumulated in both the vasculature and the epidermis in these more mature tissues (*Figure 3D and E*) although SoPIN1 seemed more abundant in the epidermis than AtPIN1 (see arrows) (SoPIN1 - 5 samples from 2 events).

In order to determine whether there were similar tissue-level differences in protein accumulation in mature tissues where AtPIN1 is implicated in branch control, we imaged AtPIN1, SoPIN1 and PIN1b in the basal internode in mature plants 1 cm above the rosette. Here AtPIN1 normally accumulates in a highly polar manner in the rootward plasma membranes of cambium (ca) and xylem parenchyma (xp) vascular-associated tissues (*Figure 3G and J*) (*Bennett et al., 2016*; *Gälweiler et al., 1998*). PIN1b accumulated in a similar pattern to AtPIN1 (*Figure 3I and L*. 10 samples from 5 events). In contrast, in addition to accumulating in the cambium and xylem parenchyma, SoPIN1 accumulated in the mature cortex (co) and central pith tissues (p) (*Figure 3H and K*. 15 samples from 4 events). AtPIN1 is not normally observed in the mature cortex or pith tissues (*Figure 3G and J*) (*Bennett et al., 2016*; *Gälweiler et al., 1998*). However we detected abundant AtPIN1 expression in the immature pith closer to the apex (*Figure 3D* box, *Figure 3—figure supplement 1* panel D) suggesting that *proAtPIN1* initially drives expression in a broad domain and that AtPIN1 and PIN1b are both cleared from the cortex and pith by maturity, while SoPIN1 is not. In the basal internode all three proteins showed the characteristic rootward polarization pattern regardless of tissue-level abundance (*Figure 3—figure supplement 1* panels D, E, F arrows).

Taken together these results show that even under the same transcriptional control AtPIN1, SoPIN1 and PIN1b show distinct tissue-level accumulation patterns in Arabidopsis. While the overall behavior of the two Brachypodium proteins is similar to AtPIN1 in many tissues, there are behaviors

unique to each. PIN1b fails to accumulate in epidermal tissues where AtPIN1 and SoPIN1 remain high, whereas SoPIN1 accumulates in the mature cortex and pith tissue where AtPIN1 and PIN1b do not. The convergent polarization patterns of SoPIN1 and the vascular accumulation of PIN1b in Arabidopsis are remarkably similar to their native behaviors in Brachypodium (*O'Connor et al., 2014*) suggesting protein-intrinsic features might control tissue-level accumulation in the two species.

## SoPIN1 but not PIN1b can restore organ initiation and bulk auxin transport in AtPIN1 null mutants

To determine whether the observed differences in SoPIN1 and PIN1b polarization and accumulation have functional consequences in Arabidopsis, we used the *proAtPIN1*-driven SoPIN1 and PIN1b constructs to complement the Arabidopsis *pin1-613* mutant (also known as *pin1-7*). The *pin1-613* allele is a putative null T-DNA insertion allele with severe organ initiation defects in the inflorescence (*Zourelidou et al., 2014*; *Bennett et al., 2006*; *Smith et al., 2006*). Given that epidermal AtPIN1 function is important for organ initiation (*Bilsborough et al., 2011*; *Kierzkowski et al., 2013*), as expected only SoPIN1 and not PIN1b was able to complement the *pin1-613* mutation and mediate organ initiation (*Figure 4A*). 3 out of 6 independent SoPIN1-expressing transgenic events showed complementation whereas all 10 independent PIN1b-expressing transgenic events failed to complement. However phenotypic complementation of *pin1-613* by SoPIN1 was incomplete with mature plants showing a variety of phenotypic defects (*Figure 4A*, *Figure 4—figure supplement 1*). Most notably each flower produced more sepals and petals than wild-type but almost no stamens (*Figure 4C*, *Figure 4—figure supplement 2*). SoPIN1-complemented *pin1-613* plants were thus sterile. We wondered whether these phenotypes could be explained by poor auxin transport function of SoPIN1 in Arabidopsis. However SoPIN1 restored wild-type levels of bulk auxin transport to *pin1-613* basal internodes (*Figure 4D*). Thus SoPIN1 is capable of supporting organ initiation and mediating rootward auxin transport in the stem, but it is not functionally identical to AtPIN1 expressed under the same promoter.

In SoPIN1-complemented *pin1-613* mutants, SoPIN1 protein accumulation in the meristem epidermis was higher than that observed in a wild-type or heterozygous genetic background and the pronounced convergent polarization patterns observed in the wild-type background were less defined (*Figure 5A*, *Figure 5—figure supplement 1*) (16 of 16 meristems). SoPIN1-complemented meristems showed a variety of phyllotactic defects and had highly variable morphologies (*Figure 5—figure supplement 1* panel B) (16 of 16 meristems). Similar to the pattern observed in the wild-type background, sub-epidermal SoPIN1 in *pin1-613* mutants accumulated in a loosely defined ring within which individual vein traces were difficult to discern (*Figure 5I*) (13 of 16 meristems). In mature tissues SoPIN1 accumulated in the epidermis, vasculature and pith, similar to the wild-type background (*Figure 5C, E and G*).

In contrast to SoPIN1, PIN1b-expressing *pin1-613* plants had pin-formed inflorescences that were indistinguishable from *pin1-613* alone (*Figure 4A*) (all 10 expressing events failed to complement). The lack of complementation mediated by PIN1b was not caused by silencing or low expression level because abundant PIN1b signal was observed in *pin1-613* meristems (23 of 26 *pin1-613* meristems from 7 events). In contrast to the wild-type background, most PIN1b expressing *pin1-613* samples had abundant epidermal expression forming a ring-shaped domain around the meristem apex (*Figure 5B and D* arrow, *Figure 5—figure supplement 2*) (14 of 19 meristems from 6 events). Also unlike the wild-type background, PIN1b in the epidermis of *pin1-613* meristems was more consistently targeted to the membrane and was often polar (*Figure 5K*). However even with this elevated polar expression in the meristem epidermis, PIN1b was unable to mediate organ initiation in *pin1-613* mutants. Below the apex PIN1b was polarized rootward in *pin1-613* meristems (*Figure 5J*), forming defined traces associated with the vasculature (*Figure 5F and L*). In the basal stem of *pin1-613* mutants PIN1b accumulated in a pattern similar to wild-type, although the arrangement of vascular bundles was irregular (*Figure 5H*). Remarkably, despite clear polar PIN1b expression in *pin1-613* mutant stems (*Figure 5M*), PIN1b was unable to rescue bulk auxin transport in this tissue (*Figure 4D*).

Because PIN1b seemed to form more defined sub-epidermal traces than SoPIN1 (Compare *Figure 3—figure supplement 1* panels B and C, and *Figure 5I and L*) we thought PIN1b combined with SoPIN1 may improve the partial SoPIN1-mediated complementation of *pin1-613*. We tested two independent PIN1b events for complementation of *pin1-613* when combined with a SoPIN1

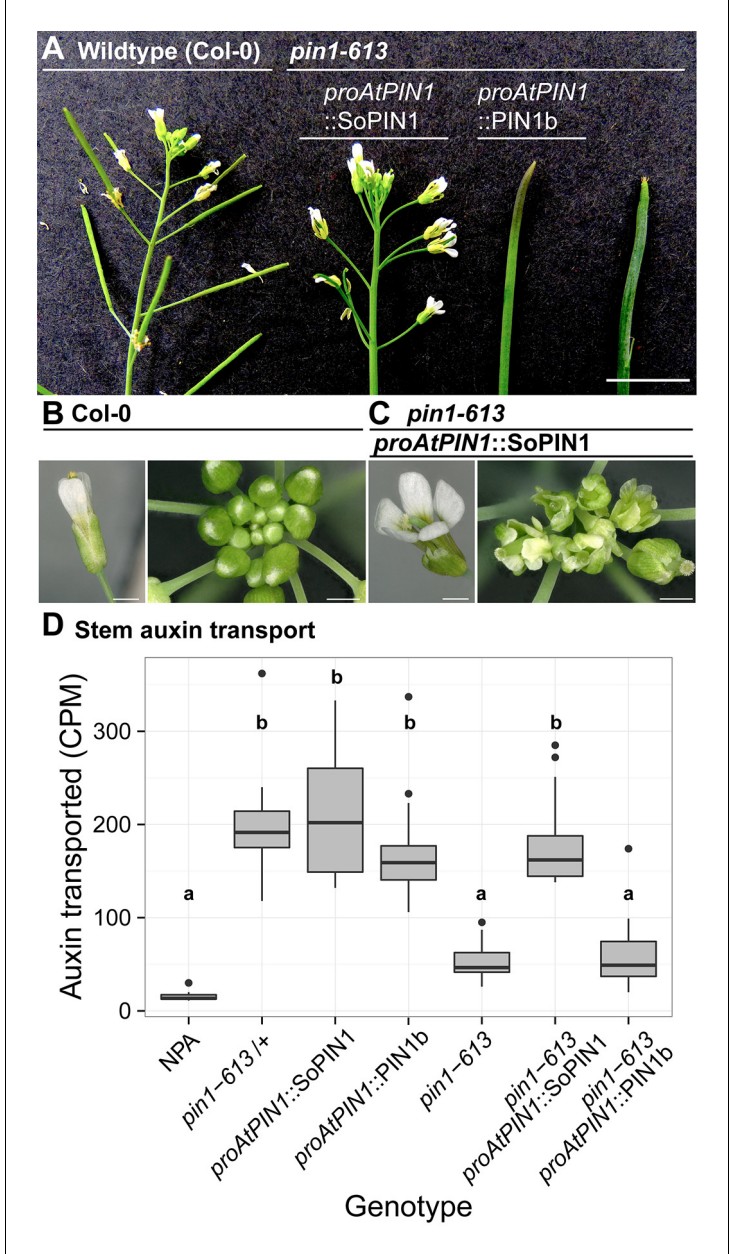

**Figure 4.** SoPIN1 but not PIN1b can partially complement the Arabidopsis *pin1-613* mutant organ initiation and bulk transport defects. (A) From left to right, inflorescence phenotypes of WT (Col-0), *proAtPIN1::SoPIN1* in *pin1-613*, *proAtPIN1::PIN1b* in *pin1-613*, and *pin1-613* alone. Note that PIN1b-expressing *pin1-613* plants are indistinguishable from *pin1-613* alone. See *Figure 4—figure supplement 1* for whole-plant phenotypes. (B) Flower (left), and inflorescence apex (right) of WT (Col-0). (C) Flower (left), and inflorescence apex (right) of *proAtPIN1::SoPIN1* complemented *pin1-613* mutants. Note the increase in petal number and lack of stamens in the flower, see *Figure 4—figure supplement 2* for organ counts. (D) Box-plot of bulk auxin transport (counts per minute, CPM) through basal internodes 1 cm above the rosette of 40-day-old Arabidopsis inflorescence stems. (n = 16 each genotype). Samples with different letters are significantly different from each other (ANOVA, Tukey HSD, p<0.05). See *Figure 4—source data 1* for source data. Scale bars: 1 cm in (A), 1 mm in (B–C).
DOI: https://doi.org/10.7554/eLife.31804.013

The following source data and figure supplements are available for figure 4:

**Source data 1.** Source data for *Figure 4D* auxin transport assays.
DOI: https://doi.org/10.7554/eLife.31804.016

**Source data 2.** Source data for *Figure 4—figure supplement 2* floral organ numbers.
*Figure 4 continued on next page*

*Figure 4 continued*

DOI: https://doi.org/10.7554/eLife.31804.017

**Figure supplement 1.** Whole-plant phenotypes of *proAtPIN1*-driven complementation of *pin1-613*.

DOI: https://doi.org/10.7554/eLife.31804.014

**Figure supplement 2.** Floral organ number in *proAtPIN1::SoPIN1* complemented flowers.

DOI: https://doi.org/10.7554/eLife.31804.015

event that showed partial complementation, but all double SoPIN1/PIN1b expressing *pin1-613* plants appeared phenotypically similar to the SoPIN1-only complementation (data not shown). Thus SoPIN1 combined with PIN1b is no better than SoPIN1 alone. In total these results demonstrate that when expressed in *Arabidopsis,* there is a clear functional separation between SoPIN1 and PIN1b independent of transcriptional control.

## SoPIN1 and PIN1b show different behaviors when expressed in the meristem epidermis

Epidermal-specific AtPIN1 expression is sufficient to rescue organ initiation in *atpin1* mutants (*Bilsborough et al., 2011*), highlighting the importance of AtPIN1 epidermal expression to organ initiation. We wanted to test specifically the ability of SoPIN1 and PIN1b to perform this epidermal function. In order to drive increased PIN1b expression in the epidermis and to help reduce transgene position-effect variation of expression level, we utilized a two-component expression system in the Landsberg *erecta* (L*er*) background to drive SoPIN1 and PIN1b under the control of the epidermis-enriched Arabidopsis ML1 promoter (Hereafter designated *proAtML1>>*) (*Lenhard and Laux, 2003*; *Sessions et al., 1999*). Under the control of *proAtML1* we achieved consistently high epidermal accumulation of both SoPIN1 and PIN1b, but similar to the *proAtPIN1* driven expression described above, only SoPIN1 showed clear convergent polarization patterns around the sites of organ initiation (*Figure 6A–6D*, *Figure 6—figure supplements 1* and *2*) (11 of 11 meristems from 2 events). Despite consistently high epidermal expression with this system, PIN1b polarity remained difficult to determine and in many cells the abundance of protein on the membrane remained low (*Figure 6D*) (13 of 13 meristems from 2 events). Instead PIN1b accumulated in intracellular bodies, especially in the cells of the apical dome and the central domain of initiating organs (*Figure 6—figure supplement 3* panels A-D). Intracellular PIN1b did not co-localize with early endosomes as assayed by FM4-64 (*Figure 6—figure supplement 3* panel C arrows), or show the perinuclear localization characteristic of the endoplasmic reticulum, suggesting accumulation in either the golgi apparatus or in vacuoles. PIN1b abundance and polarity was highest at the boundaries of lateral organs (*Figure 6—figure supplement 2*). Thus SoPIN1 and PIN1b show consistent behaviors in the meristem epidermis when expressed under either *proAtPIN1* or *proAtML1*. Despite increased PIN1b expression under *proAtML1* and a resulting increase in protein accumulation in the apex, PIN1b was still unable to form convergent polarization patterns in wild-type plants.

## Both SoPIN1 and PIN1b can rescue the Arabidopsis *pin1-4* mutation when expressed in the meristem epidermis

In order to determine whether the increased PIN1b abundance in the epidermis achieved by the *proAtML1* two-component system had functional consequences, we crossed these transgenes to the *pin1-4* mutant allele. The *pin1-4* allele is in the Landsberg *erecta* (L*er*) background and has a single P579 to L amino acid change in the second-to-last transmembrane domain of AtPIN1 (*Bennett et al., 1995*), but the phenotype is similarly severe to the null *pin1-613* allele (*Figure 7A–C*). However using immuno-localization we detected abundant AtPIN1 protein produced in *pin1-4* meristems (*Figure 7C*), while similar to previous authors we did not detect any protein in pin meristems of the null *pin1-613* allele (*Figure 7B*) (*Zourelidou et al., 2014*; *Bennett et al., 2006*) (6 *pin1-613* meristems, 5 *pin1-4* meristems). The AtPIN1 protein detected in *pin1-4* mutants accumulated primarily in the provascular tissues below the pin apex (*Figure 7C*) and appeared apolar (*Figure 7D*). Instead the perinuclear AtPIN1 localization in *pin1-4* suggests accumulation in the endoplasmic reticulum (*Figure 7D*, arrow). The presence of AtPIN1 protein produced in *pin1-4*

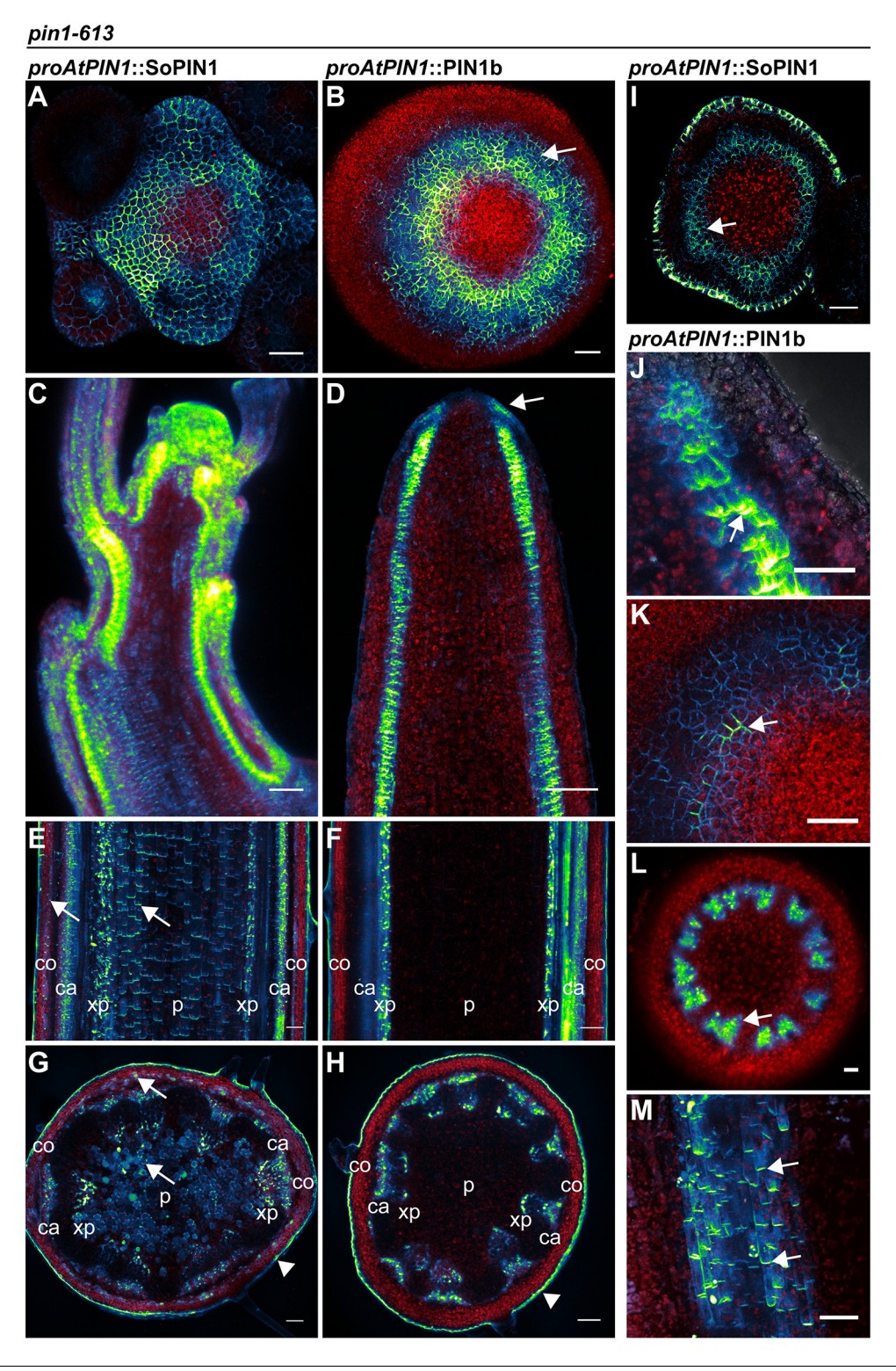

**Figure 5.** SoPIN1 and PIN1b localization in null *pin1-613* mutants. Arabidopsis *PIN1* promoter (*proAtPIN1*) driven expression of Citrine-tagged (YFP derivative) SoPIN1 and PIN1b in null *pin1-613* mutant tissue. (A,C,E,G,I) SoPIN1, (B,D,F,H,J,K,L,M) PIN1b. (A–B) Maximum projections of inflorescence meristem apexes. Arrow in (B) indicates PIN1b ring shaped epidermal domain. See *Figure 5—figure supplement 1* for SoPIN1 expression in a *pin1-613* segregating family. See *Figure 5—figure supplement 2* for more examples of PIN1b expression in *pin1-613*

*Figure 5 continued on next page*

*Figure 5 continued*

apexes. (**C–D**) Tiled confocal maximum projections of longitudinal hand-sections through inflorescence apexes. Arrow in (**D**) indicates increased PIN1b in the epidermis in the *pin1-613* background. (**E–F**) Tiled maximum projections of longitudinal hand-sections through mature basal inflorescence stem internodes 1 cm above the rosette. (**G–H**) Tiled maximum projections of hand cross-sections through mature basal internodes 1 cm above the rosette. Signal at the edge of each section (arrowheads) is cuticle auto-florescence. The cortex (co), cambium (ca), xylem parenchyma (xp), and pith (p) tissues are indicated in (**E–H**). Arrows in (**E**) and (**G**) indicate cortex and pith accumulation of SoPIN1. (**I**) Confocal z-section of SoPIN1 accumulation in a ring-shaped domain just below the apex of a complemented *pin1-613* meristem. (**J**) Longitudinal hand-section of PIN1b just below a *pin1-613* meristem apex. Arrow shows rootward polarized PIN1b. (**K**) Detail of polarized PIN1b in the meristem epidermis of a *pin1-613* meristem apex. (**L**) Cross-section of PIN1b (arrow) in distinct bundles 2 mm below a *pin1-613* meristem apex. (**M**) Rootward polarization of PIN1b (arrow) 3–4 mm below the apex of a *pin1-613* meristem. Red signal in all panels is chlorophyll auto-florescence. Scale bars: 25 µm in (**A–B**), 100 µm in (**C–H**), and 25 µm in (**I–M**).
DOI: https://doi.org/10.7554/eLife.31804.018

The following figure supplements are available for figure 5:

**Figure supplement 1.** *proAtPIN1::SoPIN1* expression in *pin1-613* segregating family.
DOI: https://doi.org/10.7554/eLife.31804.019

**Figure supplement 2.** *proAtPIN1::PIN1b* expression in *pin1-613* apexes.
DOI: https://doi.org/10.7554/eLife.31804.020

---

mutants indicates that *AtPIN1* in this background may retain partial function despite the severity of the mutant phenotype.

Indeed both SoPIN1 and PIN1b driven by *proAtML1* were able to rescue the organ formation defects of *pin1-4* (*Figure 8A*). In contrast to the partial SoPIN1-mediated complementation and failure of PIN1b to complement *pin1-613* described above, both SoPIN1 and PIN1b-complemented *pin1-4* plants made WT flowers that produced seed (*Figure 8—figure supplement 1*). In addition SoPIN1 and PIN1b were both able to rescue bulk auxin transport in the *pin1-4* basal internode, although PIN1b was less effective than SoPIN1 (*Figure 8B*). In general SoPIN1 and PIN1b-mediated complementation of *pin1-4* was phenotypically similar, but perhaps as a result of the decreased transport rate in PIN1b-complemented *pin1-4* plants, this genotype showed a significant increase in stem diameter (*Figure 8C*), providing further evidence that SoPIN1 and PIN1b are not functionally equal.

SoPIN1-complemented *pin1-4* meristems were slightly smaller than wild-type (*Figure 6—figure supplement 1*), but the protein localization was similar to the pattern observed in the WT background, with clear convergent polarization around initiating organs (*Figure 6E and G*) (10 of 10 meristems from 1 event). In contrast, compared to the WT background, PIN1b localization in *pin1-4* was dramatically altered (compare *Figure 6B* with *Figure 6F*). Most obvious was an increase in membrane targeted PIN1b and a corresponding reduction in intracellular PIN1b (*Figure 6H*, *Figure 6—figure supplement 3* panels E-H). PIN1b polarization in the *pin1-4* background was more apparent than in wild-type, and convergent polarization patterns clearly marked incipient organs (*Figure 6H*) (10 of 10 meristems from 1 event). PIN1b-complemented meristems accumulated less PIN protein in the apical dome compared to SoPIN1-complemented meristems, and the meristems were larger (*Figure 6—figure supplement 2*).

In the basal internode, both PINs had similar accumulation patterns in the cortex (co) and epidermis layers (*Figure 6I–J* arrows) and both showed rootward polarization in the epidermis (*Figure 6K–L* arrows). Despite this expression domain being drastically different than the wild-type vascular-associated pattern of AtPIN1 (*Bennett et al., 2006*; *Gälweiler et al., 1998*), expression in these few cortex layers and epidermis was apparently sufficient to drive near wild-type levels of rootward bulk auxin transport in *pin1-4* (*Figure 8B*). Thus while both proteins can complement the *pin1-4* organ initiation phenotype, the SoPIN1 and PIN1b complemented lines have differing localization patterns, slightly different auxin transport properties and minor differences in meristem and mature plant morphologies, suggesting once again that SoPIN1 and PIN1b are not functionally identical.

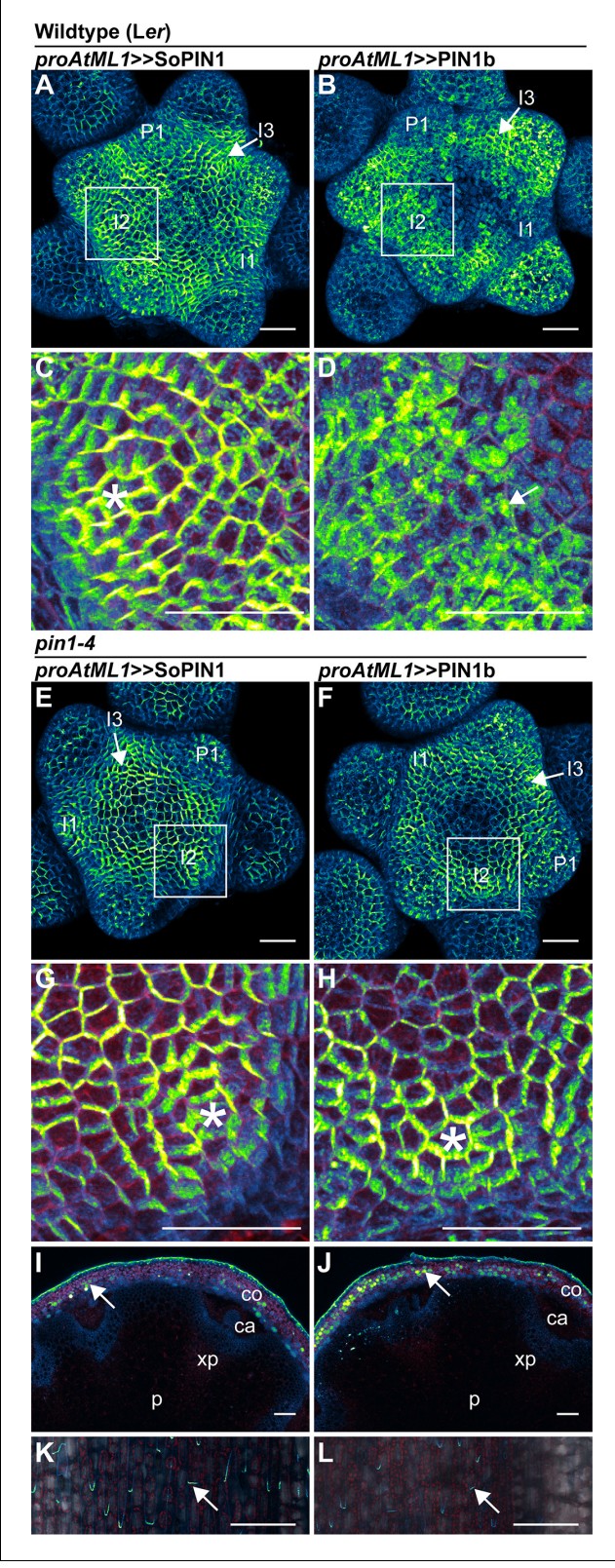

**Figure 6.** SoPIN1 and PIN1b show different behaviors under *proAtML1*-driven expression.  Maximum projections of *proAtML1::LhG4* driving pOP::SoPIN1 or pOP::PIN1b (*proAtML1 >>SoPIN1* and *proAtML1 >>PIN1b*) in wild-type Landsberg *erecta* (L*er*) (**A–D**), and *pin1-4* (**E–L**) inflorescence meristems and mature basal internodes. (**A**) SoPIN1 and (**B**) PIN1b maximum projections of wild-type L*er* inflorescence meristems. I3, I2, I1, and P1 primordia
*Figure 6 continued on next page*

*Figure 6 continued*

are indicated. White boxes around each I2 primordium indicate the regions detailed in (**C–D**). Asterisk in (**C**) indicates convergence point. Arrow in (**D**) indicates punctate PIN1b. (**E**) SoPIN1 and (**F**) PIN1b maximum projections of complemented *pin1-4* inflorescence meristems. I3, I2, I1, and P1 primordia are indicated. White boxes around each I2 primordia indicate the regions detailed in (**G–H**). Asterisks mark convergence points in (**G**) and (**H**). Red signal in (**C,D,G,H**) is cell wall propidium iodide staining. See *Figure 6—figure supplement 1* for additional samples of *proAtML1* >>SoPIN1 and *Figure 6—figure supplement 2* for additional samples of *proAtML1* >>PIN1b in both WT and *pin1-4* meristems. See *Figure 6—figure supplement 3* for details of PIN1b epidermal intracellular localization in WT and *pin1-4* meristem apexes. (**I–J**) Tiled maximum projections of cross hand-sections of mature basal internodes of SoPIN1 (**I**) and PIN1b (**J**) -complemented *pin1-4* plants showing PIN signal in the outer cortex layers (arrows). The cortex (co), cambium (ca), xylem parenchyma (xp), and pith (p) tissues are indicated. Red signal in (**I–J**) is chlorophyll auto-florescence. (**K–L**) Epidermal maximum projections showing rootward polarized PIN localization (arrows) in the basal internode of SoPIN1 (**K**), and PIN1b (**L**) -complemented *pin1-4* plants. Scale bars: 25 µm in (**A–H**). 100 µm in (**I–L**).

DOI: https://doi.org/10.7554/eLife.31804.021

The following figure supplements are available for figure 6:

**Figure supplement 1.** *proAtML1* >>SoPIN1 representative meristem maximum projections.

DOI: https://doi.org/10.7554/eLife.31804.022

**Figure supplement 2.** *proAtML1* >>PIN1b representative meristem maximum projections.

DOI: https://doi.org/10.7554/eLife.31804.023

**Figure supplement 3.** Subcellular localization of PIN1b in wild-type (L*er*) and *pin1-4* meristems.

DOI: https://doi.org/10.7554/eLife.31804.024

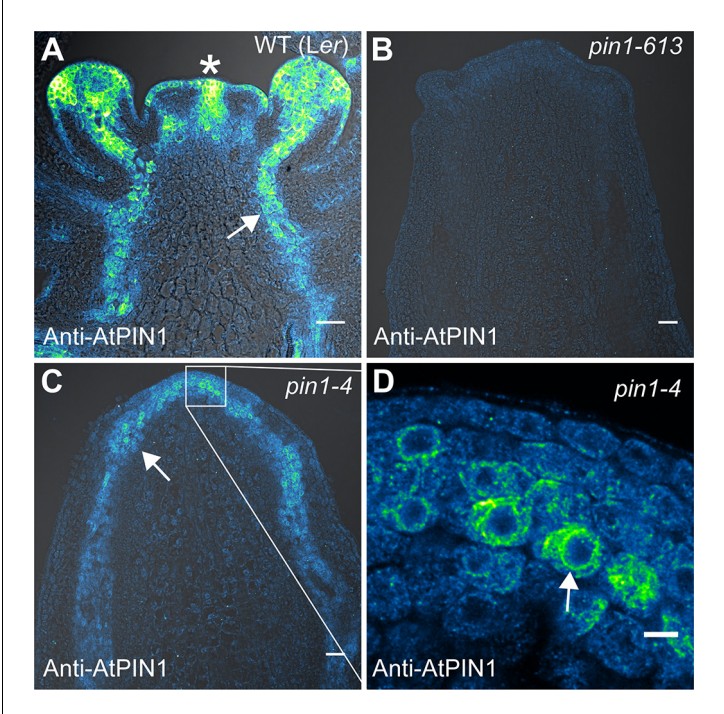

**Figure 7.** AtPIN1 protein immuno-localization in wild-type, *pin1-613,* and *pin1-4* meristems. (**A**) AtPIN1 protein accumulation in wild-type L*er* inflorescence apex shows polar PIN protein at the sites of initiating organs (asterisk), and during vein patterning below the apex (arrow). (**B**) No AtPIN1 protein is detected in *pin1-613* null mutant pin-formed apexes. (**C**) Abundant AtPIN1 protein is detected in *pin1-4* pin-formed apexes, primarily in provascular tissues below the meristem apex (arrow). Box shows region of detail in (**D**). (**D**) Detail of boxed area shown in (**C**). AtPIN1 protein in *pin1-4* accumulates in a perinuclear domain (arrow). All samples are 9 µm longitudinal sections. Scale bars: 25 µm in A-C, and 5 µm in D.

DOI: https://doi.org/10.7554/eLife.31804.025

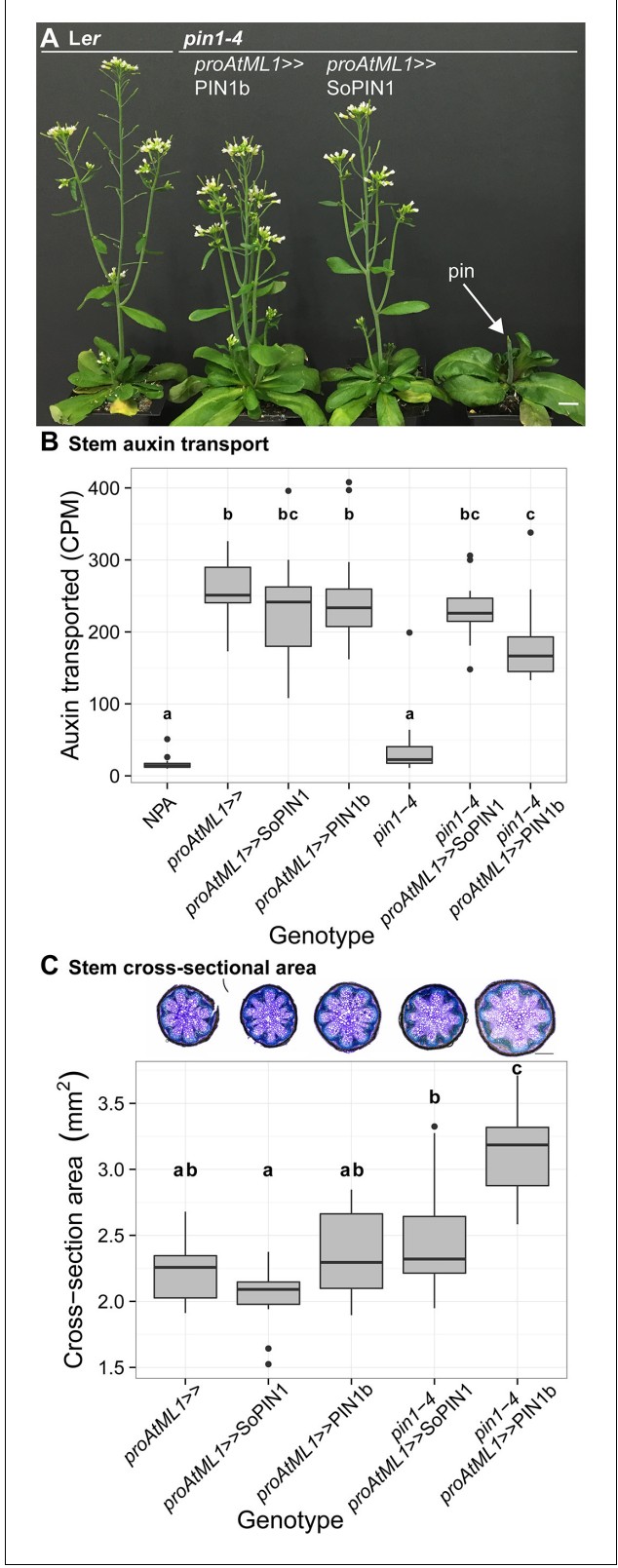

**Figure 8.** Both SoPIN1 and PIN1b can complement Arabidopsis *pin1-4* under *proAtML1*-driven expression. (**A**) From left to right, wild-type L*er*, *proAtML1* >>PIN1b complemented *pin1-4*, *proAtML1* >>SoPIN1 complemented *pin1-4*, and *pin1-4* alone. Arrow indicates barren pin inflorescence in *pin1-4*. See **Figure 8—figure supplement 1** for inflorescence phenotypes. (**B**) Box-plot of bulk auxin transport (counts per minute, CPM) through basal

*Figure 8 continued on next page*

*Figure 8 continued*

internodes 1 cm above the rosette of 40-day-old Arabidopsis inflorescence stems (n = 16 each genotype). Samples with different letters are significantly different from each other (ANOVA, Tukey HSD, p<0.05). See *Figure 8— source data 1* for source data. (C) Box-plot of stem cross-sectional area (square mm) of the mature basal internode 1 cm above the rosette (n = 12 each genotype). Samples with different letters are significantly different from each other. (ANOVA, Tukey HSD, p<0.05). See *Figure 8—source data 2* for source data. Representative Toluidine Blue O stained hand cross-sections are shown above each box for each genotype. Scale bars: 1 cm in (A). 500 µm in (C).

DOI: https://doi.org/10.7554/eLife.31804.026

The following source data and figure supplement are available for figure 8:

**Source data 1.** Source data for *Figure 8B* auxin transport assays.
DOI: https://doi.org/10.7554/eLife.31804.028

**Source data 2.** Source data for *Figure 8C* stem cross-sectional area measurements.
DOI: https://doi.org/10.7554/eLife.31804.029

**Figure supplement 1.** *proAtML1* >>SoPIN1 and *proAtML1* >>PIN1b complemented *pin1-4* inflorescence phenotypes.
DOI: https://doi.org/10.7554/eLife.31804.027

## Discussion

### The SoPIN1 and PIN1 clades have different functions in Brachypodium

During spikelet development in Brachypodium, SoPIN1 forms convergent polarization patterns surrounding the sites of organ initiation and strong expression of the auxin response reporter DR5 (*O'Connor et al., 2014*). We provide additional evidence here that SoPIN1 polarizes towards sites of high auxin concentration by showing that a DII minimum occurs at SoPIN1 convergence points. In *sopin1* mutants the reduction of DII does not occur, suggesting that SoPIN1 functions to concentrate auxin at epidermal maxima, and similar to Arabidopsis, this is required for organ initiation in the inflorescence. The barren inflorescence phenotype of *sopin1-1* mutants and the specificity of SoPIN1 for the outer tissues and for convergent polarization patterns in Brachypodium provides further support for the idea that auxin maximum formation is necessary for organ initiation and that this is primarily mediated by convergent PIN in the meristem epidermis (*Bhatia et al., 2016*; *Jönsson et al., 2006*; *Kierzkowski et al., 2013*; *Smith et al., 2006*).

SoPIN1 clade mutants have been reported in the legume *Medicago truncatula* and in tomato (*Solanum lycopersicum*) and these mutants show pleiotropic phenotypes involving phyllotaxy, organ initiation, inflorescence branching, leaf serrations and leaf compounding, but they do not form barren pin meristems (*Martinez et al., 2016*; *Zhou et al., 2011*). These wider morphogenetic events also involve epidermal PIN convergence points and associated auxin maxima (*Barkoulas et al., 2008*; *Bilsborough et al., 2011*), suggesting a general role for SoPIN1 clade members in generating such maxima. The lack of barren pin-formed meristems in these mutants suggests that different species are variably dependent on SoPIN1-generated auxin maxima for organ initiation. Even in Brachypodium and Arabidopsis, barren meristems in *sopin1* and at*pin1* respectively are restricted to later stages of development, so organs are able to form in the absence of SoPIN1 or AtPIN1 function.

In contrast to *sopin1* mutants, loss of *pin1* clade function in Brachypodium has very little effect on organ initiation despite both PIN1a and PIN1b being expressed and polarized away from auxin maxima in developing organs (*O'Connor et al., 2014*). Auxin drainage is thought necessary for proper organ size and placement (*Bhatia et al., 2016*; *Deb et al., 2015*) but the most evident phenotype in *pin1a/pin1b* mutants is the alteration of internode length. The increased internode length in *pin1b* and severely reduced internode length in *pin1a/pin1b* double mutants provides new genetic tractability to address how PINs regulate tissue growth in the shoot independent of organ initiation, a PIN function that is experimentally inaccessible in Arabidopsis because of the initiation defects of at*pin1* mutants.

Grasses contain intercalary meristems, bands of indeterminate tissue separated from the apical meristem that are responsible for internode growth after organ initiation. Auxin dynamics in this more basal meristematic tissue may be important for controlling stem growth. Indeed loss of the ABCB1 auxin exporter in maize and Sorghum results in dwarfism associated with reduced activity of

intercalary meristems (**Knöller et al., 2010**; **Multani et al., 2003**). The role of PIN1a and PIN1b in regulating intercalary meristem growth will be an important avenue for future work, especially since plant stature has played such an important role in grass domestication.

## The properties that define PIN behavior and function

### Membrane accumulation

We used heterologous expression of SoPIN1 and PIN1b in Arabidopsis to explore the ways in which different PIN family members may have different properties post-transcription (Summarized in **Figure 9**). When expressed in the meristem epidermis in wild-type Arabidopsis, SoPIN1 is localized to the membrane in most cells while PIN1b often accumulates internally (**Figure 6—figure supplement 3**). Thus with the same transcriptional control, different PINs can vary in the degree to which, after protein production, they accumulate at the plasma membrane. The differential membrane targeting of PIN1b and SoPIN1 is a tissue-specific phenomenon however, because unlike in the epidermis, in

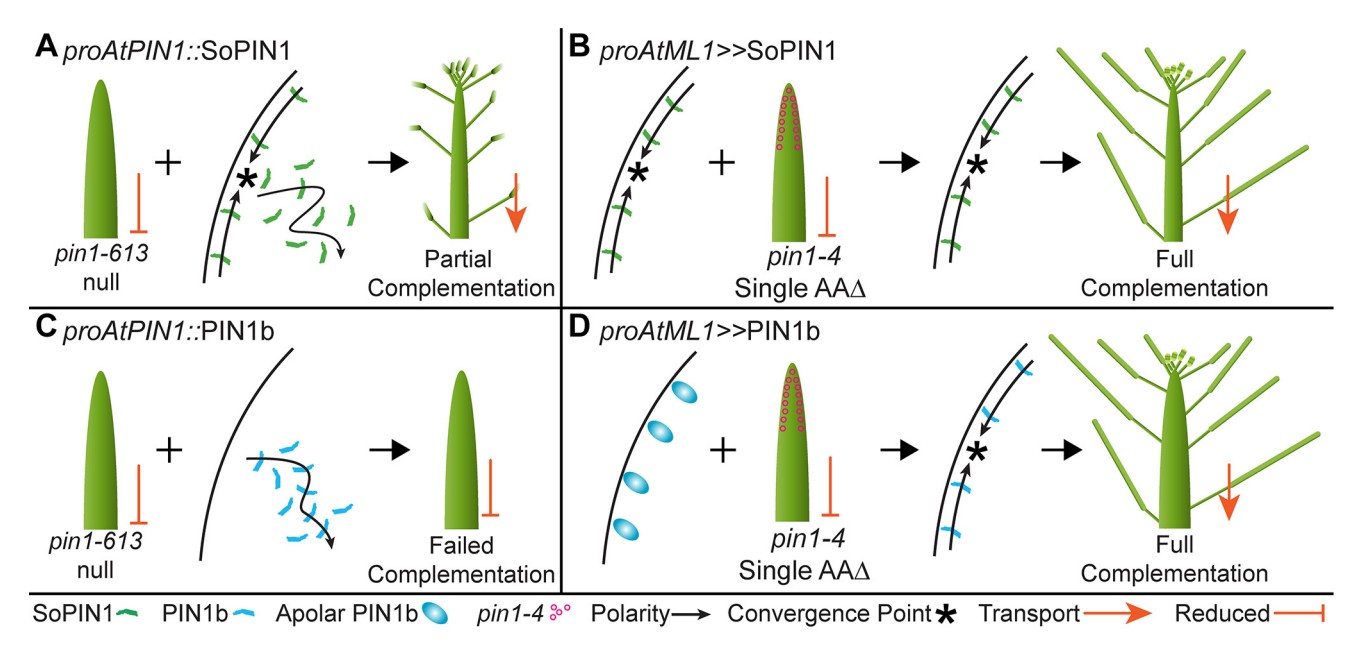

**Figure 9.** Heterologous expression visual summary: Functional distinction between PIN auxin efflux proteins during development. Polarized SoPIN1 is represented by green lines, polarized PIN1b by blue lines, un-polarized PIN1b by blue ovals, and the putative partially functional *pin1-4* protein is indicated by magenta circles. Red arrows indicate measured auxin transport in the mature basal internode, while red bar-headed lines indicated reduced transport. Black arrows represent polarized PIN patterns. Convergence points are marked with asterisks. (A) When expressed in both the epidermis and internal tissues with *proAtPIN1* in wild-type Col-0, SoPIN1 forms convergent polarization patterns in the epidermis and is partially able to rescue the organ initiation phenotypes and bulk transport in null *pin1-613* mutants. (B) When SoPIN1 is expressed in the epidermis from the *proAtML1* promoter, it forms convergence points in the wild-type L*er* background and is able to rescue more fully the organ initiation phenotypes of the *pin1-4* single amino acid change mutation. (C) In contrast, when PIN1b is expressed in both the epidermis and internal tissues from the *proAtPIN1* promoter in wild-type Col-0, it accumulates mostly in the internal tissues, and is unable to complement the *pin1-613* organ initiation phenotype. It is also unable to transport auxin through mature stem segments, despite apparently AtPIN1-like accumulation and polarization in the stem. (D) When PIN1b is expressed in the epidermis from the *proAtML1* promoter it does not form convergent polarization patterns and is often un-polarized in the wild-type L*er* background (blue ovals), but it does in the *pin1-4* background, where it is able to rescue the defective organ initiation phenotype and mediate bulk transport. See **Figure 9—figure supplement 1** for a protein alignment comparing AtPIN1 to other PIN1-clade protein sequences from diverse angiosperms.

DOI: https://doi.org/10.7554/eLife.31804.030

The following source data and figure supplement are available for figure 9:

**Source data 1.** FASTA alignment source data for **Figure 9—figure supplement 1**.
DOI: https://doi.org/10.7554/eLife.31804.032
**Figure supplement 1.** Brassicaceae-specific PIN1 domains.
DOI: https://doi.org/10.7554/eLife.31804.031

the basal internode both PINs accumulate at the rootward plasma membrane (*Figure 3—figure supplement 1* panels E and F). The regulation of PIN plasma membrane polar targeting and endocytic recycling has been an important avenue for understanding PIN function and general membrane protein biology (*Luschnig and Vert, 2014*). Our results provide further evidence that at least some of the signals governing membrane accumulation are inherent in, and vary between, different PIN family members (*Wisniewska et al., 2006*).

## Tissue accumulation

Under the same transcriptional control AtPIN1, SoPIN1 and PIN1b show different tissue-level accumulation patterns in Arabidopsis. In wild-type plants, *proAtPIN1*-expressed PIN1b shows reduced overall accumulation in the epidermis compared to AtPIN1 or SoPIN1 (Compare *Figure 3D–3F*). Even with greater expression under *proAtML1,* the intracellular PIN1b signal observed in the meristem epidermis (*Figure 6—figure supplement 3*) suggests that PIN1b protein could be actively targeted to the vacuole for degradation as has been shown for PIN2 in the root (*Abas et al., 2006*; *Kleine-Vehn et al., 2008b*). In contrast, SoPIN1 is abundant in the meristem epidermis and accumulates in the mature cortex and pith tissues where AtPIN1 and PIN1b do not (*Figure 3H* arrows). Because we observed clear AtPIN1 accumulation in the immature pith (*Figure 3—figure supplement 1* panel D), this suggests that *proAtPIN1* is at least initially active in pith tissue and that the PIN1 clade members AtPIN1 and PIN1b are removed later in development, while SoPIN1 is not. Laser capture micro-dissection and RNAseq of mature stem tissue has detected abundant *AtPIN1* transcription in the mature pith (T. Greb, personal communication, February 2017) suggesting that even at maturity AtPIN1, and probably PIN1b, are actively cleared post-transcriptionally from this tissue. Our results suggest that the AtPIN1 expression domain is far broader than is indicated by the protein accumulation pattern and further highlight the importance of PIN post-transcriptional regulation for controlling PIN tissue-level abundance.

In Arabidopsis endogenous PIN family members show a degree of cross-regulation where loss-of-function mutations in one PIN family member result in tissue-level accumulation of a different PIN in a compensatory pattern (*Blilou et al., 2005*; *Paponov et al., 2005*; *Vieten et al., 2005*). We observed similar behavior in the *pin1-613* null background where SoPIN1 and PIN1b accumulation in the meristem epidermis was increased in the absence of AtPIN1 (*Figure 5—figure supplements 1* and *2*). This sensitivity of SoPIN1 and PIN1b to the presence of AtPIN1, even while under the *AtPIN1* promoter, suggests that wild-type AtPIN1 is able to compete with SoPIN1 and PIN1b post-transcriptionally for residency at the membrane. However we did not observe the same competitive effect with *proAtML1* where SoPIN1 and PIN1b tissue-level accumulation, though not membrane residence, seemed similar between *pin1-4* mutant and wild-type meristems (*Figure 6—figure supplements 1* and *2*). This difference between *proAtPIN1* and *proAtML1* may be because PIN transcription under *proAtPIN1* is sensitive to the dosage of other PINs as has been suggested (*Vieten et al., 2005*). Alternatively, the lack of increased SoPIN1 and PIN1b accumulation in the *pin1-4* mutant background may be because the AtPIN1 protein produced in *pin1-4* is still able to compete with the other expressed PINs. Regardless, these results highlight the sensitivity of PIN tissue-level abundance to both transcriptional and post-transcriptional control, and the variability between PIN family members independent of transcription.

## Transport activity

In Arabidopsis phosphorylation of PINs by several different families of protein kinases is thought necessary for efficient auxin transport (*Zourelidou et al., 2014*; *Barbosa et al., 2014*; *Jia et al., 2016*; *Willige et al., 2013*). PIN activation by phosphorylation may explain the inability of PIN1b to mediate bulk auxin transport in the basal internode of *pin1-613* plants despite being expressed, accumulating at the membrane and being polarized rootward in this tissue (*Figure 4D*, *Figure 5M*). It is possible that in the *proAtPIN1* domain PIN1b does not interact with the appropriate activating kinase and it is thus unphosphorylated and inactive. Indeed a partially unphosphorylatable form of AtPIN1 fails to complement fully the bulk auxin transport defect of *pin1-613* mutants in the basal internode (*Zourelidou et al., 2014*) and AtPIN1 polarity can be uncoupled from phosphorylation status, and thus presumably transport activity can be independent of polarization (*Weller et al., 2017*).

However, when expressed using *proAtML1*, PIN1b expression in the outer tissue layers of the basal internode appears sufficient to mediate bulk auxin transport in *pin1-4* (*Figure 8B*). One explanation for this is that PIN1b activity may be tissue dependent, perhaps because of the differing expression domains of activating kinases (*Zourelidou et al., 2014*). Arabidopsis PIN4 and PIN7 are present in the *proAtML1* domain (*Bennett et al., 2016*) making it conceivable that these PINs are the normal targets of activating kinases in this tissue. Alternatively, kinases in the L*er* genetic background may be more effective at activating PIN1b than those in Col, but the dramatic effect of the *pin1-4* allele vs wild-type *AtPIN1* on PIN1b polarization behavior within the L*er* background makes this explanation unlikely (see below). In either case the behavior of PIN1b in *pin1-613* provides a clear indication that even once a PIN has accumulated at the cell membrane in a tissue it may not be active.

## Interaction

A particularly striking result is the ability of PIN1b to form convergent polarization patterns and mediate organ initiation in the *pin1-4* missense mutant background when it is unable to do so in the null *pin1-613* background. It is unlikely that differences between *proAtPIN1* and *proAtML1*-mediated expression can explain this differential complementation because both promoters drive expression in the epidermis and both promoters are sufficient to complement *atpin1* mutants using wild-type AtPIN1 as well as SoPIN1. As described above, it is possible that differences in activating enzymes or similar interactors between the L*er* and Col backgrounds could contribute to the strikingly different behavior of PIN1b in *pin1-4* vs *pin1-613*. Indeed mutation of the leucine-rich repeat receptor-like kinase *ERECTA,* which is mutated in the L*er* background, has known effects on PIN1 localization when combined with other mutations in the *ERECTA* family (*Chen et al., 2013*; *Torii et al., 1996*). However the dramatic effect of wild-type AtPIN1 vs *pin1-4* on PIN1b membrane targeting (compare *Figure 6—figure supplement 3* panels A-D to E-H) within the L*er* background suggests the differing genetic backgrounds of each complementation (L*er* vs Col-0) is not sufficient to explain the differential complementation.

Instead the strong influence of *pin1-4* on PIN1b membrane targeting and polarity in the meristem epidermis suggests that PIN1b may be cooperating with a partially functional *pin1-4* protein and together they recapitulate the organ initiation functions of wild-type AtPIN1. PIN1b interaction with *pin1-4* in the outer cortex of the stem may also explain the ability of PIN1b to rescue bulk transport in the basal internodes of *pin1-4* mutants while it cannot in the null *pin1-613* allele. Partial *pin1-4* function is further supported by the result that SoPIN1 complementation of the null *pin1-613* allele is incomplete and because of flower defects the plants are sterile (*Figure 4C*), while SoPIN1-mediated complementation of *pin1-4* is complete and flowers are phenotypically normal and set seed (*Figure 8—figure supplement 1*). Accordingly SoPIN1 convergent polarization patterns are more evident in the presence of *pin1-4* than they are during complementation of the null *pin1-613* allele (Compare 5A and 6E), further evidence that residual *pin1-4* function augments SoPIN1 during *pin1-4* complementation. Combined these data suggest that the *pin1-4* allele is hypomorphic and that it provides some necessary function to PIN1b.

If PIN1b is indeed inactive in null *pin1-613* mutants as we hypothesized above then it is possible *pin1-4* facilitates the interaction of PIN1b with the appropriate activating kinase and this allows PIN1b to perform organ initiation and bulk transport. Alternatively interaction between PIN1b and *pin1-4* may facilitate proper membrane targeting or polarization of either protein, resulting in functional transport. The increased level of polar, plasma-membrane localized PIN1b in *pin1-4* meristems supports the idea that *pin1-4* controls PIN1b membrane residency, but it cannot explain why PIN1b appears unable to mediate bulk transport in *pin1-613* despite being membrane-localized and polar in the basal internode.

Direct PIN-PIN interactions have so far not been shown, but if one PIN type can convey targeting, polarity or activity information to another through direct or indirect interaction this may be relevant to auxin transport in tissues where multiple PINs are coexpressed, such as in the Arabidopsis root meristem (*Blilou et al., 2005*), or in the shoot apical meristems of most angiosperms where the SoPIN1 and PIN1 clade proteins likely overlap, as they do during spikelet development in Brachypodium (*O'Connor et al., 2014*).

## Polarity

We previously showed that the polarization dynamics of SoPIN1, PIN1a and PIN1b in Brachypodium could be modeled by assigning two different polarization modes to the SoPIN1 and PIN1 clades (*O'Connor et al., 2014*). In the model, SoPIN1 orients toward the adjacent cell with the highest auxin concentration thus transporting auxin up the concentration gradient and providing a positive feedback to concentrate auxin into local maxima. In contrast, in the model PIN1a and PIN1b proteins are allocated in proportion to net auxin flux thus providing a positive feedback in which flux through the tissue is amplified by the allocation of PIN1a/b in the direction of that flux. The assignment of two different polarization modes was previously used to describe the behavior of AtPIN1 during organ placement and vein patterning using an auxin-concentration based switching mechanism between the up-the-gradient (UTG) and with-the-flux (WTF) polarization modes (*Bayer et al., 2009*). However it has also been suggested that a flux-based mechanism alone can account for both convergence points and vein patterning (*Abley et al., 2016*; *Cieslak et al., 2015*; *Stoma et al., 2008*).

Despite evidence that convergent PIN polarization is dependent on localized auxin signaling in adjacent cells (*Bhatia et al., 2016*), there are still no proven mechanisms for direct sensing of intercellular auxin gradients or auxin fluxes across membranes. The *sopin1*, *pin1a* and *pin1b* phenotypes in Brachypodium are consistent with different polarization modes. SoPIN1 is required for organ initiation and the formation of auxin maxima in Brachypodium, which is primarily modeled using UTG polarization (*Bayer et al., 2009*; *Jönsson et al., 2006*; *Smith et al., 2006*). On the other hand, *pin1a* and *pin1b* mutant plants do not show organ initiation defects, but rather only have internode elongation defects, a tissue where WTF models have been used to explain PIN dynamics and measured auxin transport kinetics during vein patterning and the regulation of branch outgrowth (*Bayer et al., 2009*; *Bennett et al., 2016*; *Mitchison, 1980*, *1981*; *Prusinkiewicz et al., 2009*).

In wild-type Brachypodium, the SoPIN1 and PIN1a/b expression domains are almost entirely mutually exclusive (*O'Connor et al., 2014*), making it possible that the observed polarization differences between the two clades are due to expression context or tissue-level stability and not to functional differences between the proteins themselves. More specifically, perhaps an UTG mechanism dominates the epidermis while a WTF mechanism is utilized in the internal tissues and different PINs interact equally with these context-dependent mechanisms. Our heterologous expression studies do not exclusively support context-dependent or protein-dependent mechanisms for SoPIN1 and PIN1 polarization. It is clear that alone only SoPIN1 and AtPIN1 show the convergent polarization patterns associated with UTG polarization, and alone only SoPIN1 and AtPIN1 are thus able to mediate organ initiation, while PIN1b cannot. On the other hand, all three PINs are capable of rootward polarization in the basal internode tissue and PIN1b can be co-opted to convergent polarization at the meristem epidermis in the presence of *pin1-4*. The results presented here do not demonstrate whether within a single cell SoPIN1 and PIN1b would orient differently with respect to auxin as might be expected for the dual polarization model (*O'Connor et al., 2014*). However such context-independent polarization behavior was previously observed for PIN1 and PIN2 in the root where both PINs can polarize in opposing directions within a single cell type when expressed in the PIN2 domain (*Kleine-Vehn et al., 2008a*; *Wisniewska et al., 2006*).

## Outlook

In total our Brachypodium mutant phenotypes and heterologous expression results point to multiple levels at which PIN family members can be functionally distinct. Differential membrane targeting, tissuelevel accumulation, transport activity, indirect or direct interaction and the resultant polarity may all contribute to the dynamics of PIN action during plant development. In most flowering plants two PIN clades, SoPIN1 and PIN1, with differing functions and differing transcriptional and post-transcriptional properties mediate auxin transport in the shoot, but these properties are seemingly combined into AtPIN1 in Arabidopsis and other Brassicaceae species. Because PIN1b alone is unable to mediate organ initiation while AtPIN1 can, and these two PINs are both members of the same clade, AtPIN1 may have gained the ability to form convergent polarization patterns and mediate organ initiation after or coincident with the loss of the SoPIN1 clade. Indeed when comparing Brassicaceae PIN1 proteins against a broad sampling of other angiosperm PIN1 proteins, the Brassicaceae PIN1 proteins have several divergent protein domains (*Figure 9—figure supplement 1*), suggesting possible neofunctionalization within the Brassicacea family. Alternatively an expansion of the PIN3,4,7

clade is also characteristic of Brassicaceae species (*Bennett et al., 2014*; *O'Connor et al., 2014*), making it possible duplicated members of this clade buffered the loss of SoPIN1. However there is no indication that PIN3,4,7 have a role in organ initiation in the inflorescence (*Guenot et al., 2012*). Regardless, we believe the combination of SoPIN1 and PIN1 characteristics into AtPIN1 coincident with the loss of the SoPIN1 clade represents a form of reverse-subfunctionalization, the combination of functions originally split between homologs into a single protein after gene loss. It is not surprising that PINs may be particularly amenable to this kind of functional evolution because, as described above, there are several post-transcriptional regulatory steps that ultimately combine to control PIN function in plants. The output of auxin transport is the sum of an extensive network of post-transcriptional interactions that all act to regulate auxin transport itself, and this gives the system plasticity during development and perhaps also over evolutionary time.

# Materials and methods

## *sopin1-1*, *pin1a-1*, and *pin1b-1* creation with CRISPR

SoPIN1 (Bradi4g26300), PIN1a (Bradi1g45020) and PIN1b (Bradi3g59520) were targeted with CRISPR using vectors developed for rice (*Miao et al., 2013*). CRISPR constructs were transformed into Brachypodium inbred line Bd21-3 using previously published methods (*Bragg et al., 2015*).

### *sopin1-1* CRISPR

The SoPIN1 guide was AGGCTGTCGTACGAGGAGT. This guide was shorter than the typical 20 bp in an effort to provide greater target specificity for SoPIN1 (*Fu et al., 2014*). In the T0 regenerated plants 5 out of 9 independent transgenic events showed severe organ initiation defects and all 5 contained lesions in the SoPIN1 CRIPSR target site. Unfortunately only one of the events with a T0 phenotype set seed. In the T1 progeny of this event only those individuals that contained the CRISPR transgene showed lesions in the SoPIN1 CRISPR target site and these plants showed the *sopin1* phenotype and thus failed to set seed, suggesting active editing by the SoPIN1 CRISPR transgene in this event.

Not all events showed such efficient editing however, and we identified an independent T1 family where a C insertion in the SoPIN1 CRISPR target site co-segregated with the barren inflorescence phenotype. We designated this allele, which causes a premature stop codon before the end of the third exon codon 739 base pairs downstream from the target site, *sopin1-1*. (Primer IDs 1–2 *Table 1*) We backcrossed a heterozygous *sopin1-1* plant to the Bd21-3 parental line and all F1 progeny (N = 4) were wild-type. In the F2 generation, plants homozygous for the *sopin1-1* lesion co-segregated with the barren inflorescence phenotype (N = 60: 32 het, 18 homo,10 wt). Amongst these plants, 16 did not have the Cas9 transgene (Primer IDs 3–4 *Table 1*) and the barren inflorescence phenotype still co-segregated with the *sopin1-1* lesion (N = 16: 8 het, 3 homo, 5 wt). We crossed the T1 *sopin1-1* heterozygous plant with a line homozygous for the previously published SoPIN1-Citrine genomic reporter line (*O'Connor et al., 2014*). In the F2 we identified two individuals homozygous for *sopin1-1* but heterozygous for the SoPIN1-Citrine transgene. Only F3 progeny individuals that lacked the SoPIN1-Citrine transgene showed a *sopin1-1* phenotype, while those that contained the SoPIN1-Citrine transgene made spikelets and set seed (N = 34: 6 *sopin1-1* phenotype, 28 wt phenotype) (*Figure 1—figure supplement 1*). This complementation was independent of the presence of Cas9.

### *pin1a-1* CRISPR

The PIN1a guide was ATCTACTCCCGGCGGTCCAT. We identified edited plants in the T1 generation (Primer IDs 5–6 *Table 1*), then found a homozygous T insertion in the T2 generation which was independent of Cas9, resulting in a premature stop codon 939 base pairs downstream, here designated *pin1a-1*. No single-mutant *pin1a-1* phenotypes were observed.

### *pin1b-1* CRISPR

The PIN1b guide was AGGGCAAGTACCAGATCC. We identified a single plant from the regenerating T0 PIN1b CRISPR population that had longer basal internodes and twisted leaves. This plant was homozygous for an A deletion in the PIN1b CRISPR target site causing a premature stop in the

**Table 1.** Primers
See methods for usage.

| ID# | Name | Sequence | Purpose |
|---|---|---|---|
| 1 | 524_Bradi4g26300_4230_F | CGTTCCGTGTTGATTCCGATG | *sopin1-1* genotyping with HgaI digestion |
| 2 | 525_Bradi4g26300_4923_R | CTGGAGTAGGTGTTGGGGTTC | *sopin1-1* genotyping with HgaI digestion |
| 3 | 526_Cas9_8622_F | TCCCAGAGAAGTACAAGGAGATCT | Cas9 Genotyping |
| 4 | 527_Cas9_9159_R | TTGTACACGGTGAAGTACTCGTAG | Cas9 Genotyping |
| 5 | 104_BdPIN_11_QPCR_F | ACAACCCTTACGCCATGAAC | *pin1a-1* genotyping with NcoI digestion |
| 6 | 473_PIN1a_dom1_shortR | CACACGAACATGTGCAGGTC | *pin1a-1* genotyping with NcoI digestion |
| 7 | 541_Bradi3g59520_PIN1b_5084_F | TGATGCTCTTCATGTTCGAGTACC | *pin1b-1* genotyping with mboI digestion |
| 8 | 542_Bradi3g59520_PIN1b_5838_R | GGAGTAAACTACGTTGTGACAAGG | *pin1b-1* genotyping with mboI digestion |
| 9 | 019 - Ubi-1 Prom attB4 F | GGGGACAACTTTGTATAGAAAAGTTGCTGCAGTGCAGCGTGACCCGG | pZmUbi amplification for cloning |
| 10 | 020 - Ubi-1 Prom attB1 R | GGGGACTGCTTTTTTGTACAAACTTGCTGCAGAAGTAACACCAAACA | pZmUbi amplification for cloning |
| 11 | PIN1pro-GW-F | GGGGACAACTTTGTATAGAAAAGTTGTTACCCTCATCCATCATTAACTT | *proAtPIN1* amplification |
| 12 | PIN1pro-GW-R | GGGGACTGCTTTTTTGTACAAACTTGTCTTTTGTTCGCCGGAGAAGAGA | *proAtPIN1* amplification |
| 13 | 455 BdSoPIN1 cacc mRNA | TCACATCTGCTGCCGCTGCC | SoPIN1-Citrine coding region amplification |
| 14 | 302 - PIN_7 qPCR UTR R2 | AATCCCAAAAGCCGACATTG | SoPIN1-Citrine coding region amplification |
| 15 | 466 BdPIN1b cacc mRNA-2 | CACCTGTACACACTGCGGCGCT | PIN1b-Citrine coding region amplification |
| 16 | 308 - PIN_5 qPCR UTR R1 | ACTCGCTAACCAACCCCTTAATT | PIN1b-Citrine coding region amplification |
| 17 | MVR087 - pin1-613 RP (SALK_047613) | AATCATCACAGCCACTGATCC | *pin1-613* genotyping |
| 18 | MVR086 - pin1-613 LP (SALK_047613) | CAAAAACACCCCCAAAATTTC | *pin1-613* genotyping |
| 19 | MVR036 - LBb1.3 | ATTTTGCCGATTTCGGAAC | *pin1-613* genotyping |
| 20 | 344 - Citrine Seq R | GAAGCACATCAGGCCGTAG | PIN1b-Citrine and SoPIN1-Citrine genotyping |
| 21 | 524_Bradi4g26300_4230_F | CGTTCCGTGTTGATTCCGATG | SoPIN1-Citrine genotyping |
| 22 | 541_Bradi3g59520_PIN1b_5084_F | TGATGCTCTTCATGTTCGAGTACC | PIN1b-Citrine genotyping |
| 23 | 543_pin1-4_Aci_F | GCTTTTGCGGCGGCTATGAGATTTGT | *pin1-4* genotyping with AciI digestion |
| 24 | 544_pin1-4_Aci_R | GCTTCTGATTTAATTTGTGGGTTTTCA | *pin1-4* genotyping with AciI digestion |
| 25 | 076 - BASTA_F2 | CTTCAGCAGGTGGGTGTAGAG | ML1::LhG4 genotyping |
| 26 | 077 - BASTA_R2 | GAGACAAGCACGGTCAACTTC | ML1::LhG4 genotyping |

DOI: https://doi.org/10.7554/eLife.31804.033

second exon 502 base pairs downstream, here designated *pin1b-1* (Primer IDs 7–8 **Table 1**). All T1 progeny showed the *pin1b* phenotype and were homozygous for the *pin1b-1* lesion. We back-crossed these T1 plants to Bd21-3 and all F1 progeny had a wild-type phenotype (N = 11). In the F2, the *pin1b* phenotype co-segregated with the *pin1b-1* lesion (N = 215, 91 het, 39 homo, 26 wt). Amongst these plants, 24 did not have the Cas9 transgene and the *pin1b* phenotype still co-segregated perfectly with the *pin1b-1* lesion (N = 24: 10 het, six homo, eight wt). We crossed *pin1b-1* without Cas9 to a line homozygous for the previously published PIN1b-Citrine transgene

(*O'Connor et al., 2014*). In the F3 we identified lines homozygous for both the transgene and *pin1b-1* and used these to quantify internode lengths compared to *pin1b-1* (*Figure 2—figure supplement 1*).

### *pin1a-1/pin1b-1* double mutant

Homozygous *pin1b-1* lacking Cas9 was crossed to homozygous *pin1a-1* lacking Cas9. In the F2 phenotypically *pin1b-1* plants that were also genotyped heterozygous for *pin1a-1* were identified. In the homozygous *pin1b-1* F3 generation the double *pin1a-1/pin1b-1* mutant phenotype segregated perfectly with the *pin1a-1* lesion (N = 23: 10 het, five homo, eight wt). Double *pin1a-1/pin1b-1* mutants were easily identified by phenotype and produce seed.

## Brachypodium reporter constructs

All constructs were cloned using Multi-site Gateway (Invitrogen Grand Island, NY) and were transformed into Brachypodium Bd21-3 using previously published methods (*Bragg et al., 2015*). For pZmUbi::DII-Venus, we first cloned the maize ubiquitin promoter into pDONR P4-P1R (Primer IDs 9–10 *Table 1*) and this was subsequently recombined with pDONR 221 containing Arabidopsis DII and pDONR P2R-P3 containing Venus-N7 (*Brunoud et al., 2012*) into the Multi-site Gateway binary vector pH7m34GW (http://gateway.psb.ugent.be/). In the T3 generation degradation of DII-Venus in the presence of auxin was validated by treating excised Brachypodium spikelet meristems with 1 µM 1-naphthaleneacetic acid (NAA) or mock treatment in 70% ethanol and imaging every 30 min (*Figure 1—figure supplement 2*).

For SoPIN1-Cerulean, the promoter plus 5' coding pDONR-P4-P1R and 3' coding plus downstream pDONR-P2R-P3 fragments from (*O'Connor et al., 2014*) were used. Maize codon-optimized Cerulean fluorescent protein, courtesy of David Jackson, was amplified with 5x Ala linkers and cloned into pENTR/D-TOPO. These three fragments were then recombined into pH7m34GW.

## Arabidopsis reporter constructs

All constructs were cloned using Multi-site Gateway (Invitrogen) and transformed using standard floral dip. For *proAtPIN1* complementation, a 3.5 kb Arabidopsis PIN1 promoter region was amplified from a genomic clone previously reported to complement the *pin1* (*Heisler et al., 2005*) and cloned into Gateway vector pDONR P4-P1R (Primer IDs 11–12 *Table 1*). For each Brachypodium PIN-Citrine fusion construct, the entire PIN coding region, including the Citrine insertion, was amplified from the previously published reporter constructs (*O'Connor et al., 2014*) and cloned into pENTR/D TOPO (Primer IDs 13–16 *Table 1*). The Citrine fusion in each is located in a position known to complement *pin1* mutations (*Heisler et al., 2010*; *Wisniewska et al., 2006*; *Xu et al., 2006*). The *proAtPIN1* pDONR P4-P1R and PIN coding region pENTR/D-TOPO vectors were then recombined into Gateway binary vector pH7m24GW (http://gateway.psb.ugent.be/) and transformed by floral dip into both Col-0 and plants heterozygous for *pin1-613* (also known as *pin1-7*, SALK_047613) (*Bennett et al., 2006*; *Smith et al., 2006*). Complementation was assessed in the T3 generation, and all plants were genotyped for both the *pin1-613* mutation (Primer IDs 17–19 *Table 1*) and for presence of the PIN transgene (Primer IDs 20–22 *Table 1*).

For the *proAtML1* lines the PIN coding regions with Citrine insertion pENTR/D TOPO Gateway vectors were recombined downstream of the two-component OP promoter in vector pMoA34-OP (*Moore et al., 1998*) and then transformed into the *proAtML1* driver line in the Landsberg *erecta* background (*Lenhard and Laux, 2003*). Lines homozygous for both the *proAtML1* driver and OP:: PIN were crossed to het *pin1-4* and complementation was assessed in the F2 and F3 generations. All complemented plants were genotyped for *pin1-4* (Primer IDs 23–24 *Table 1*), the Brachypodium PINs (Primer IDs 20–22 *Table 1*) and the presence of the ML1 driver transgene (Primer IDs 25–26 *Table 1*).

## Confocal and Scanning Electron Microscopy

All confocal images were captured on a Zeiss 780 laser scanning confocal using a W Plan-Apochromat 20x or 63x magnification 1.0 numerical aperture objectives (Zeiss, Jena, Germany). Detection wavelengths: 517–570 nm for Citrine-tagged PINs, 535–552 for DII-Venus, 463–509 for SoPIN1-CERULEAN, 490–543 for AtPIN1-GFP, 691–753 nm for FM4-64, 561–606 nm for Dylight 549, 631–

717 nm for Propidium Iodide and 646–726 for chlorophyll A auto-fluorescence. The pinhole was set to one airy unit for all meristem stacks and details of sub-epidermal polarization but was open to the maximum setting for tiled longitudinal and cross sections of the basal internode (*Figures 3D–L*, *5C–H and* and *6I–J*). Detection gain and laser power were varied according to signal strength unless direct comparisons between genotypes were made as indicated in figure legends, except for *Figure 1J and K* where the DII gain was higher in *Figure 1K* in order to show DII distribution.

Cryo scanning electron microscopy was performed on a Zeiss EVO HD15 SEM fitted with a LaB6 filament and a Quorum PP3010T (Quorum Technololgies, Lewes, Sussex. UK) cryo preparation unit using the BSD (Backscattered electron) detector with probe current as set to 10 nA, and 15.00 kV gun voltage. Frozen samples were coated in <1.5 nm Pt.

### Auxin transport assays

Auxin transport assays were carried out as described in (*Crawford et al., 2010*). Briefly, 17 mm long basal internodes were excised and the apical end submerged in 30 μl Arabidopsis salts (ATS) without sucrose (pH = 5.6) containing 1 μM 14C-IAA (American Radiolabeled Chemicals, ST. Louis, M ). After 6 hr incubation the basal 5 mm segment was excised, cut in half, and shaken overnight at 400 RPM in 200 μl scintillation liquid prior to scintillation counting. 10 μM N-1-Naphthylphthalamic Acid (NPA), an auxin transport inhibitor, was added prior to incubation for negative controls.

### AtPIN1 Immuno-Localization

Detection of AtPIN1 in sectioned apexes was performed as previously described (*O'Connor et al., 2014*). Commercial polyclonal goat anti-AtPIN1 (AP-20) was used at a concentration of 1:150 (Santa Cruz Biotechnology, Dallas, Texas). Affinity-purified Donkey Anti-Goat Dylight 549 secondary was used at a concentration of 1:200 (Jackson Immuno Research, West Grove, PA, USA). Control samples where the primary antibody was omitted showed a similar level of background signal as *pin1-613* null mutant samples.

## Acknowledgements

Thanks to Raymond Wightman for SEM assistance, Martin van Rongen for assistance with transport assays and *pin1-613* oligos, Tom Bennett for *proAtPIN1* oligos, Marcus Heisler for AtPIN1-GFP construct and *pin1-4* genotyping assistance, Teva Vernoux for DII plasmids, David Jackson for maize codon-optimized Cerulean and to all the members of the Leyser lab. Thanks also to Graeme Mitchison, Katie Abley, Michael Raissig, and Pau Formosa-Jordan for helpful comments on the manuscript.

## Additional information

### Funding

| Funder | Grant reference number | Author |
|---|---|---|
| Gatsby Charitable Foundation | | Devin Lee O'Connor<br>Samuel Elton<br>Fabrizio Ticchiarelli<br>Ottoline Leyser |
| US Department of Energy | DE-AI02-07ER64452 | Mon Mandy Hsia<br>John P. Vogel |

The funders had no role in study design, data collection and interpretation, or the decision to submit the work for publication.

### Author contributions

Devin Lee O'Connor, Conceptualization, Resources, Data curation, Formal analysis, Supervision, Validation, Investigation, Visualization, Methodology, Writing—original draft, Project administration, Writing—review and editing; Samuel Elton, Fabrizio Ticchiarelli, Mon Mandy Hsia, Investigation; John P Vogel, Resources, Supervision, Funding acquisition; Ottoline Leyser, Conceptualization,

Resources, Supervision, Funding acquisition, Methodology, Project administration, Writing—review and editing

## Author ORCIDs

Devin Lee O'Connor http://orcid.org/0000-0003-4071-8626
Samuel Elton http://orcid.org/0000-0003-4470-4758
Fabrizio Ticchiarelli http://orcid.org/0000-0001-5744-2393
John P Vogel http://orcid.org/0000-0003-1786-2689
Ottoline Leyser http://orcid.org/0000-0003-2161-3829

## Decision letter and Author response

Decision letter https://doi.org/10.7554/eLife.31804.035
Author response https://doi.org/10.7554/eLife.31804.036

## Additional files

### Supplementary files
• Transparent reporting form
DOI: https://doi.org/10.7554/eLife.31804.034

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
