## [Decision Letter]

[Editors’ note: a previous version of this study was rejected after peer review, but the authors submitted for reconsideration. The first decision letter after peer review is shown below.]

Thank you for submitting your work entitled "Cross-species functional diversity within the PIN auxin efflux protein family" for consideration by *eLife*. Your article has been evaluated by a Senior Editor and three reviewers, one of whom, Dominique C Bergmann (Reviewer #1), is a member of our Board of Reviewing Editors. The following individual involved in review of your submission has agreed to reveal their identity: Dolf Weijers (Reviewer #2).

Our decision has been reached after consultation among the reviewers. Based on these discussions and the individual reviews below, we regret to inform you that your work will not be considered further for publication in *eLife*.

While all three reviewers through the topic was important and interesting, all felt that the key conclusions made in this study required additional experimental support (detailed in the comments below, but focused on what could be said with the heterologous expression system, concerns about interpretations of rescue of the different *pin1* alleles, and feelings that better imaging was required to make statements about different BdPIN subcellular localization). The reviewers and editors do want to encourage you to submit this manuscript again once these key experiments, all of which should be technically feasible, have been done.

Reviewer #1:

In Arabidopsis, the single PIN1 protein has many roles. One fascinating, but confusing element of PIN1 polarization and auxin transport is that they are sometimes acting in "with the flux" way, and sometime "up the gradient". Making the critical observation that the Brassicas are unusual in having lost the sister of PIN1 (SoPIN1) clade, but that this is retained in grasses, O'Conner and colleagues test whether these two functions ascribed to Arabidopsis PIN1 might be separable in another species. This work is a nice follow up from modeling papers by the same authors and introduce what is potentially a powerful system for future studies. I liked where this was going, but found there were several areas where the conclusion outstripped the data and that different experiments could give them much more power to address the interesting questions. Major issues 1-4 are listed below.

1) Choice of Arabidopsis rather than native context has some limitations. To address the question of differential functions/duplications, this is obviously the right choice. However, when it comes time to discuss what these Brachy proteins do, interpretations must be more cautious. Given the capacity shown by this group and others to make transgenics and CRISPRs, more analysis in Brachypodium should be possible and would enable the authors to be able to say much more about the divergence in function between the Brachy proteins.

2) The AtPIN1pro-driven expression of the Brachy proteins showed clear differences between them, but the relationship between these patterns and AtPIN1pro was unclear. Is the broad expression of soPIN1 because the AtPIN1 promoter is actually expressed here and AtPIN1 is regulated at the protein level to be less? Or are there sequences in soPIN1 (genomic?) that are influencing expression in the heterologous system?

3) The relevance of the non-polar localization of *Pin1b* is not clear; is it aggregating or endosomal (i.e., does this represent a regulated difference)? The images are not clear enough to tell. In general, given the imaging capacities at SLCU and the fact that these studies were done in Arabidopsis, the image quality in close-ups should be higher.

4) The portion about differences in rescuing *pin1-613* vs. *pin1-4* is potentially interesting, but it is critical to ensure that it is the specific allele of *pin1*, not the absence of ERECTA that is responsible for difference in behavior. Second, does PIN1-4-GFP actually behave as predicted in the model? Can it be differentially recruited?

Reviewer #2:

O'Connor et al. describe a thorough analysis of the differences in function between two Brachypodium PIN proteins: SoPIN1 and PIN1b. Most research on PIn proteins has been performed in Arabidopsis, that O'Conner et al. previous showed to lack a specific subclade of PIN proteins, named SoPIN1. It was also previously shown that SoPIN1 and PIN1b have different localization patterns in Brachypodium, but it remained an open question whether there was a functional difference between the two proteins. Here, the authors generate CRISPR/Cas9 mutants in the SoPIN1 and PIN1b genes and show that SoPIN1 appears to represent the functional ortholog of Arabidopsis PIN1, while PIN1b plays a different, and more subtle role in development. Next, by using Arabidopsis wild-type and two different *pin1* mutant alleles as a model, the authors show that the two proteins also behave differently when heterologously expressed. Again, SoPIN1 behaves more like AtPIN1, while BdPIN1b has different properties. This is in line with the finding that SoPIN1 is more effective at complementing a null allele of AtPIN1 than PIN1b is. The authors present several speculations on the basis for these differences.

All in all, the work is sound and provides an important perspective on diversification of PIN1-related proteins, as well as the evolutionary paths of this function. There are several issues that need to be clarified or elaborated upon:

1) The comparison of SoPIN1 and PIN1B in Brachypodium could have been developed further. Is the difference based on protein properties or does the pattern and level of expression contribute to different function? The authors raise this question, which can in fact be addressed in Brachyposium, but move to Arabidopsis to address it. The *pin1-4* complementation assay in Arabidopsis suggests that both proteins can perform the same function in at least this context. It would be helpful if the authors complemented the *sopin1* mutant with PIN1B driven by the SoPIN1 regulatory sequences.

2) The differential accumulation of SoPIN1 and PIN1b in Arabidopsis requires more explanation. First, the statements about subcellular localization could be better supported by high-magnification images. Second, the accumulation of SoPIN1 in the pith tissue is remarkable. Is the AtPIN1 promoter expressed there? Is this due to differential protein stability or are there cis-elements for expression regulation in the SoPIN1 coding sequence (as was shown for MP in Marcus Heisler's lab; Bhatia et al., Curr. Biol. 2016)?

3) The authors convincingly show that the two Brachypodium PIN proteins behave differently in Arabidopsis when expressed from the same promoter, but is somewhat unsatisfying that there are no indications as to the nature of these differences. Given that genetic materials are readily available, it would be good if the authors tested whether the different localization patterns are associated with altered trafficking or degradation. For both these properties, standardized drug treatments (BFA, wortmannin, MG132, etc.) are established, and even roots could be used since the proteins are expressed from the AtPIN1 promoter.

4) The interpretation of the complementation experiment in the *pin1-4* background is somewhat problematic. It is presented as a distinction between a null allele (the Col T-DNA insertion; *pin1-613*) and a missense allele (*pin1-4*). However, since the *pin1-4* allele is in a Landsberg background, an alternative explanation is that there are ecotype-specific modifiers that affect that ability of the different PIN1 homologues to act in Arabidopsis. This should at least be discussed as an alternative explanation.

Reviewer #3:

This manuscript by O'Connor and colleagues explores the functionality of proteins from two clades of PIN1-related transporters present in Brachypodium, PIN1 and SoPIN1, to deepen our understanding of PIN polarization processes and PIN-regulated patterning at the shoot apical meristem. SoPIN1 has been lost in Brassicacea and some of the authors (including the first authors) previously proposed that proteins form the two clades had different properties in their capacity to polarize in brachypodium while the unique PIN1 from Brassicacea would have evolved to have all the range of properties and/or functions shown by the two ancestral proteins. The manuscript includes a significant amount of work describing mutants in PIN1b and SoPIN1 in Brachypodium and the polarization patterns and phenotypes obtained upon heterologous expression in Arabidopsis. From the different phenotypes of the Brachypodium mutants and from the different capacities of the proteins to complement Arabidopsis *pin1* mutants, the authors conclude that 1) the two types of PIN1-related proteins have different functions and 2) there might be functional differences although their Discussion clearly highlights the fact that their data are far from being conclusive on this aspect of the work.

There are in my view two major problems in this work:

1) As SoPIN1 and PIN1a and b have different patterns of expression, it is not such of a surprise that they would have different functions but showing it properly using mutant of Brachypodium would clearly be a strength of the manuscript. However only mutants for PIN1b have been obtained and the authors justify this by saying that PIN1a "accumulates at the site of vein formation after the organs begins to grow". Not only is this a limited justification for an absence of redundancy between these two close homologs of PIN1 but this statement surprisingly misrepresents published data from some of the authors showing that PINa is likely co-expressed with PINb in the vein before any sign of organ outgrowth (Figure 2 in O'Connor et al. PLoS Comp Biol 2014 and corresponding text stating that PINa and PIN1b are expressed starting from i1). Thus, the authors cannot conclude from the single *pin1b* mutant that "PIN1b is expendable for organ initiation" and this weakens the rest of the study.

2) The heterologous expression of PIN1b and SoPIN1 in Arabidopsis is an interesting strategy but also a dangerous one as little is known on the mechanisms of polarity in Arabidopsis and it is also not known whether any of the knowledge from Arabidopsis is valid in other species. The authors use translational version of the proteins with Citrine and nothing is said about the site of Citrine insertion. The site of insertion of a fluorescent protein is known to have the capacity to affect polarity of PINs (Wisniewska et al. Science 2006) and nothing tells us that the polarization capacities of PIN1b and SoPIN1 translational fusions are not affected by the insertion. Proving it would require proof that they polarize normally in Brachypodium and/or that they complement the phenotype. Along the same line, overexpression of PIN1 can affect its polarity (e.g. Figure 4 of Benkova et al. Cell 2003). This could explain some of the results observed with loss of polarity of PIN1b and SoPIN1. It could perhaps also explain why PIN1b is less polar – and also seen in what could be vacuoles – (Figure 5) only when the endogenous PIN1 is also present. This and also the absence of knowledge on the stability of the proteins are some of the reasons that make the results presented very difficult to analyze. There are a few other reasons such as the number of transgenics that is somehow low (e.g. 7 lines expressing PIN1b under AtPIN1 promoter is not sufficient to conclude that it cannot complement *pin1-6*, especially when it can complement *pin1-4*). At the end and despite the amount of works it represents, the manuscript does not really bring new knowledge to the very complex problem of how PINs get polarized. Going beyond the simplest conclusion that is that PINb and SoPIN1 might have different activities and cellular properties seem difficult with the data presented. And even this conclusion is not entirely convincing when one consider the data from Figure 5 that show very nice polarization patterns towards flower initia with both PIN1b and SoPIN1.

---

## [Author Response]

[Editors’ note: the author responses to the first round of peer review follow.]

Reviewer #1:[…] 1) Choice of Arabidopsis rather than native context has some limitations. To address the question of differential functions/duplications, this is obviously the right choice. However, when it comes time to discuss what these Brachy proteins do, interpretations must be more cautious. Given the capacity shown by this group and others to make transgenics and CRISPRs, more analysis in Brachypodium should be possible and would enable the authors to be able to say much more about the divergence in function between the Brachy proteins.

We agree that more can be done in Brachypodium. We added *pin1a* single and *pin1a/b* double mutant analyses, which provide a more complete genetic dissection of the SoPIN1 and PIN1 clades in Brachypodium. Along with these additional mutant genotypes we have provided SEMs of meristem defects (Figure 1) and improved quantification of mutant phenotypes (Figure 1 and Figure 2). We have also added Brachypodium complementation experiments that show the previously published native reporters can complement the single-mutant phenotypes of *sopin1* and *pin1b* (Figure 1—figure supplement 1 and Figure 2—figure supplement 1). Combined we believe these experiments provide a greatly improved analysis of the native Brachypodium protein functions.

2) The AtPIN1pro-driven expression of the Brachy proteins showed clear differences between them, but the relationship between these patterns and AtPIN1pro was unclear. Is the broad expression of soPIN1 because the AtPIN1 promoter is actually expressed here and AtPIN1 is regulated at the protein level to be less? Or are there sequences in soPIN1 (genomic?) that are influencing expression in the heterologous system?

We have provided additional imaging of the native AtPIN1 reporter that shows early protein accumulation in pith tissue, suggesting that *proAtPIN1* is at least initially active in the pith (Figure 3, Figure 3—figure supplement 1). Laser-capture micro-dissection and RNAseq performed in the lab of Thomas Greb has also detected abundant AtPIN1 transcription in the mature pith. These data suggest that *proAtPIN1* does indeed drive expression in the pith tissue, and that AtPIN1 and PIN1b, but not SoPIN1, are cleared post-transcription.

If there are additional regulatory sequences within the SoPIN1 coding region, which indeed includes introns and part of the 5’ and 3’ UTRs, we might expect these to drive SoPIN1 in the pith while under the *proAtML1* two-component system, which we do not see. In addition, when we cloned the SoPIN1 and PIN1b coding regions we designed the primers to not include 5’ and 3’ conserved non-coding sequences, which might contain regulatory information. Despite this, we cannot completely exclude the genomic regions as sources of transcriptional information.

3) The relevance of the non-polar localization of Pin1b is not clear; is it aggregating or endosomal (i.e., does this represent a regulated difference)? The images are not clear enough to tell. In general, given the imaging capacities at SLCU and the fact that these studies were done in Arabidopsis, the image quality in close-ups should be higher.

We have added additional high-resolution close-ups of PIN1b subcellular localization in wild-type and *pin1-4* meristems (Figure 6—figure supplement 3). While these images don’t necessarily provide a clear indication of where PIN1b is accumulating, they are at least consistent with PIN1b accumulation in either the vacuole or golgi, which may indicate a regulated response as has been shown for PIN2 in the Arabidopsis root.

4) The portion about differences in rescuing pin1-613 vs. pin1-4 is potentially interesting, but it is critical to ensure that it is the specific allele of pin1, not the absence of ERECTA that is responsible for difference in behavior. Second, does PIN1-4-GFP actually behave as predicted in the model? Can it be differentially recruited?

We have provided immuno-localizations showing that AtPIN1 protein accumulates in the *pin1-4* allele, but not in the null *pin1-613* allele (Figure 7). While the presence of protein in *pin1-4* does not directly address the background effects between Col and L*er,* it does provide a clear indication that the two *atpin1* alleles are different, and thus could explain the difference in PIN1b complementation. We maintain that the dramatic difference in PIN1b localization between wild-typeand *pin1-4* within the L*er* background further suggests that the *atpin1* mutant allele is the most likely explanation for the differential complementation. That said, we cannot completely discount background effects, including *erecta,* which has known effects on PIN1 localization. We have added an expanded discussion of the potential for background effects to the Discussion section (subsection “Transport activity”, last paragraph and subsection “Interaction”, first paragraph). In addition, we have begun several new experiments to directly address background effect, and *erecta,* but these will take a significant amount of additional time.

Reviewer #2:[…] There are several issues that need to be clarified or elaborated upon:1) The comparison of SoPIN1 and PIN1B in Brachypodium could have been developed further. Is the difference based on protein properties or does the pattern and level of expression contribute to different function? The authors raise this question, which can in fact be addressed in Brachyposium, but move to Arabidopsis to address it. The pin1-4 complementation assay in Arabidopsis suggests that both proteins can perform the same function in at least this context. It would be helpful if the authors complemented the sopin1 mutant with PIN1B driven by the SoPIN1 regulatory sequences.

While Brachypodium is indeed more experimentally tractable than other grass models, especially in regards to transgenics, it is far more cumbersome than Arabidopsis. We intend to continue to use both models in the future, and certainly additional experiments in Brachypodium are necessary, including cross-complementation between the clades as the reviewer suggests. The complementation of the *sopin1* and *pin1b* mutantsby their respective native reporters, which has been added to the revised manuscript (Figure 1—figure supplement 1 and Figure 2—figure supplement 1) provides important groundwork for these future experiments as it shows that the fusion proteins are at least functional in their native contexts. In the present manuscript, we felt additional genetic analysis with *pin1a* mutants was far more important, especially considering likely genetic redundancy in the PIN1 clade. We hope to be able to report more results in Brachypodium in the future.

2) The differential accumulation of SoPIN1 and PIN1b in Arabidopsis requires more explanation. First, the statements about subcellular localization could be better supported by high-magnification images. Second, the accumulation of SoPIN1 in the pith tissue is remarkable. Is the AtPIN1 promoter expressed there? Is this due to differential protein stability or are there cis-elements for expression regulation in the SoPIN1 coding sequence (as was shown for MP in Marcus Heisler's lab; Bhatia et al., Curr. Biol. 2016)?

As described above, we have provided high-magnification images of PIN1b subcellular localization in both wild-type and *pin1-4* (Figure 6—figure supplement 3).

We have also provided imaging evidence that AtPIN1 is indeed initially expressed in the pith tissues, suggesting that AtPIN1 and PIN1b are cleared post-transcriptionally while SoPIN1 is not. At this time, we cannot address the nature of what features account for this differential accumulation, but this will be an important avenue for future research.

3) The authors convincingly show that the two Brachypodium PIN proteins behave differently in Arabidopsis when expressed from the same promoter, but is somewhat unsatisfying that there are no indications as to the nature of these differences. Given that genetic materials are readily available, it would be good if the authors tested whether the different localization patterns are associated with altered trafficking or degradation. For both these properties, standardized drug treatments (BFA, wortmannin, MG132, etc.) are established, and even roots could be used since the proteins are expressed from the AtPIN1 promoter.

Most drug treatments are indeed standardized for analysis in the root. Unfortunately, we have found that the concentrations of drugs that are effective in the root are not necessarily effective in the shoot (BFA for example), making use of these drugs in the shoot context problematic. Roots could be used for these experiments as the reviewer suggests, but it would be hard to relate results from the root to the functional assays we present here which are focused on the shoot. It is indeed a limitation of the present paper that we cannot explain the nature of the differential behaviors, however, the functional analyses established in the present work will provide ample material to address these differences in the future.

4) The interpretation of the complementation experiment in the pin1-4 background is somewhat problematic. It is presented as a distinction between a null allele (the Col T-DNA insertion; pin1-613) and a missense allele (pin1-4). However, since the pin1-4 allele is in a Landsberg background, an alternative explanation is that there are ecotype-specific modifiers that affect that ability of the different PIN1 homologues to act in Arabidopsis. This should at least be discussed as an alternative explanation.

We agree that the role of genetic background cannot be completely discounted. We added an expanded discussion of possible background effects to the Discussion section (subsection “Transport activity”, last paragraph and subsection “Interaction”, first paragraph). However, as outlined in the Discussion we believe that the dramatic change in PIN1b localization within the L*er* background suggests differences in the *pin1* mutant allele as the most likely explanation for the differential complementation.

Reviewer #3:[…] There are in my view two major problems in this work:1) As SoPIN1 and PIN1a and b have different patterns of expression, it is not such of a surprise that they would have different functions but showing it properly using mutant of Brachypodium would clearly be a strength of the manuscript. However only mutants for PIN1b have been obtained and the authors justify this by saying that PIN1a "accumulates at the site of vein formation after the organs begins to grow". Not only is this a limited justification for an absence of redundancy between these two close homologs of PIN1 but this statement surprisingly misrepresents published data from some of the authors showing that PINa is likely co-expressed with PINb in the vein before any sign of organ outgrowth (Figure 2 in O'Connor et al. PLoS Comp Biol 2014 and corresponding text stating that PINa and PIN1b are expressed starting from i1). Thus, the authors cannot conclude from the single pin1b mutant that "PIN1b is expendable for organ initiation" and this weakens the rest of the study.

We completely agree that the lack of a *pin1a* mutant, and the likely genetic redundancy between PIN1a and PIN1b was a limitation of our previous manuscript. As described above we have added single *pin1a* as well as double *pin1a/b* mutant analyses (Figure 1 and Figure 2), providing a more complete analysis of SoPIN1 and PIN1 clade function in Brachypodium. The lack of an organ initiation phenotype in *pin1a/b* double mutants provides support for the idea that the SoPIN1 and PIN1 clades have substantially different roles in Brachypodium.

2) The heterologous expression of PIN1b and SoPIN1 in Arabidopsis is an interesting strategy but also a dangerous one as little is known on the mechanisms of polarity in Arabidopsis and it is also not known whether any of the knowledge from Arabidopsis is valid in other species. The authors use translational version of the proteins with Citrine and nothing is said about the site of Citrine insertion. The site of insertion of a fluorescent protein is known to have the capacity to affect polarity of PINs (Wisniewska et al. Science 2006) and nothing tells us that the polarization capacities of PIN1b and SoPIN1 translational fusions are not affected by the insertion. Proving it would require proof that they polarize normally in Brachypodium and/or that they complement the phenotype.

Indeed, proving that the Brachypodium fusion proteins are functional is an important experiment. We have added experiments where the *sopin1-1* and *pin1b-1* mutants are complemented by the previously published native reporter lines, demonstrating that the fusion lines are indeed functional. In addition, the expression and polarization dynamics observed in Brachypodium were validated by immunolocalization experiments performed using SoPIN1 and PIN1 antibodies in maize (O'Connor et al. PLoS Comp Biol 2014).

As indicated in the Materials and methods, for each Brachypodium PIN-CITRINE fusion construct, the entire PIN coding region, including the CITRINE insertion, was amplified from the previously published reporter constructs (O'Connor et al., 2014). The methods now state that the FP insertion position is the same used in AtPIN1-GFP lines already shown to functionally complement *pin1* mutants (Heisler et al., 2010; Wisniewska et al., 2006; Xu et al., 2006) (Lines 754-756).

Along the same line, overexpression of PIN1 can affect its polarity (e.g. Figure 4 of Benkova et al. Cell 2003). This could explain some of the results observed with loss of polarity of PIN1b and SoPIN1. It could perhaps also explain why PIN1b is less polar – and also seen in what could be vacuoles – (Figure 5) only when the endogenous PIN1 is also present.

This was precisely our motivation for using the two-component system, which is far less susceptible to position-effect variation of expression level. There is no doubt that expression using this system could be described as “overexpression”, but it is at least consistent between the lines that express SoPIN1 and those that express PIN1b. So, while both proteins are “overexpressed”, only PIN1b shows intracellular accumulation while SoPIN1 tends to be more stable on the membrane. This a very artificial system, but it acts to highlight the differential behaviors of the two proteins, which is our primary goal.

This and also the absence of knowledge on the stability of the proteins are some of the reasons that make the results presented very difficult to analyze. There are a few other reasons such as the number of transgenics that is somehow low (e.g. 7 lines expressing PIN1b under AtPIN1 promoter is not sufficient to conclude that it cannot complement pin1-6, especially when it can complement pin1-4).

We believe that 7 lines expressing PIN1b where we can observe abundant PIN1b protein in the pin meristem is sufficient to conclude that PIN1b cannot complement *pin1-613.* We screened additional lines for complementation, but did not perform microscopy to validate that PIN1b was indeed expressed, which is an important test.

At the end and despite the amount of works it represents, the manuscript does not really bring new knowledge to the very complex problem of how PINs get polarized. Going beyond the simplest conclusion that is that PINb and SoPIN1 might have different activities and cellular properties seem difficult with the data presented. And even this conclusion is not entirely convincing when one consider the data from Figure 5 that show very nice polarization patterns towards flower initia with both PIN1b and SoPIN1.

We believe that providing ample evidence that the SoPIN1 and PIN1 clades are functionally different is a worthy goal because it provides functional assays for future studies on how PIN polarization is controlled. We agree that this is a very complex problem, but the complexity only acts to reaffirm that subtle differences in PIN function may offer a way to understand the differential polarization dynamics between clades, and in turn how PINs respond to auxin. Our work adds complexity to an already complex problem, but it also adds new trajectories for future work.